# Conflicting adaptations in an inhibitory feedback circuit

Gregor A. Bergmann[1,2] 🆔, Melissa W. Tan[1,2] 🆔, Katie Greenin-Whitehead[1,2] 🆔, Philippe J. Fischer[1,2] 🆔, Thomas C. Cozens[1,2] 🆔 and Andrew C. Lin[1,2] 🆔

[1]*School of Biosciences, University of Sheffield, Firth Court, Western Bank, Sheffield, United Kingdom*
[2]*Neuroscience Institute, University of Sheffield, Firth Court, Western Bank, Sheffield, United Kingdom*

The peer review history is available in the Supporting Information section of this article (https://doi.org/10.1113/JP290394#support-information-section).

**Abstract figure legend** We studied activity-dependent adaptation in the fruit fly *Drosophila*'s memory centre, the mushroom body. Here, excitatory Kenyon cells (KCs) receive feedback inhibition from the anterior paired lateral (APL) neuron. When KCs are artificially overactivated for 24 h, both KCs and APL reduce their sensitivity to excitation ('↓ Exc', '↓ Na$^+$ channels'), resulting in conflicting adaptations on KCs, that is, both reduced excitation and reduced inhibition.

**Abstract** Neural networks maintain stable activity levels by compensating for perturbations through homeostatic plasticity. However, homeostatic mechanisms operating at different levels may conflict with each other. For example, in inhibitory feedback circuits, if inhibitory neurons receive excess excitation, compensation at a 'local' level (e.g. reducing inhibitory neurons' activity) could conflict with 'network-level' compensation (e.g. suppressing the excitatory neurons responsible for over-exciting the inhibitory neurons). We studied this problem in the *Drosophila* mushroom body, where excitatory Kenyon cells (KCs) receive feedback inhibition from the anterior paired lateral (APL) neuron. Dual-colour calcium imaging revealed that prolonged (24 h) artificial activation of KCs causes APL to become less sensitive to KC activity. Meanwhile, KCs compensate for their excess activity by reducing excitation, yet this change is opposed by reduced inhibition from APL. This conflict meant that KCs did not consistently show the expected homeostatic reduction in odour responses. Our findings show that neurons sometimes adapt their activity locally in a way that counteracts broader adaptations in the network.

**Gregor A. Bergmann** completed his PhD at the University of Veterinary Medicine in Hanover, Germany, where he worked on isolated neurons of the locust antennal lobe that form inhibitory circuits. He then joined the Lin lab at the University of Sheffield, where his research focused on homeostatic regulation of inhibitory circuits in *Drosophila melanogaster*. He is interested in how a homeostatic balance of excitation and inhibition is maintained during and after neuronal maturation; how perturbations of this balance influence nervous system functionality; and how it is affected by environmental factors such as xenobiotics.

(Received 27 October 2025; accepted after revision 4 February 2026; first published online 5 March 2026)
**Corresponding author** A. C. Lin: School of Biosciences, University of Sheffield, Firth Court, Western Bank, Sheffield, UK.   Email: andrew.lin@sheffield.ac.uk

**Key points**

- Neural networks maintain stable activity levels through homeostatic plasticity – but what physiological variables are stabilised?
- In inhibitory feedback circuits, local and network-level compensation might conflict. For example, if excitatory neurons are overactive, they might compensate by becoming less excitable. But if inhibitory neurons compensate for the excess excitation by also becoming less excitable, this would decrease inhibition onto the excitatory neurons and increase their activity.
- We tested this idea in the fruit fly brain, where excitatory Kenyon cells (KCs) get negative feedback from an inhibitory neuron called anterior paired lateral (APL).
- After overactivation of KCs, APL becomes less sensitive to KCs. The resulting loss of inhibition onto KCs counteracts KCs' attempts to reduce their activity.
- These results show that adaptation at local and network levels can conflict with each other.

# Introduction

Biological systems can compensate for perturbations, but sometimes compensatory mechanisms misfire, leading to counterproductive or maladaptive plasticity (Kuner & Flor, 2017; Morris & Rogers, 2013; Mullins et al., 2016; Storz & Scott, 2019; Takeuchi & Izumi, 2012; Velotta & Cheviron, 2018). Here we study this general problem using neuronal homeostatic plasticity, a phenomenon whereby neural networks stabilise levels or patterns of activity in the face of perturbations, by adjusting synaptic strengths and/or intrinsic excitability to return activity to a set point (Wen & Turrigiano, 2024). Depending on which variables are being homeostatically controlled, homeostatic changes in different parts of a network might support or oppose each other. For example, if excitatory neurons are overly active, they should decrease their excitability. In inhibitory feedback networks, this decrease can be supported by a homeostatic increase in feedback inhibition (Das et al., 2011; Kuhlman et al., 2013; Sachse et al., 2007), for example via increased excitability of inhibitory neurons (Chand et al., 2015; Gainey et al., 2018) or stronger excitatory-to-inhibitory synapses (Chang et al., 2010). However, the opposite might occur if more 'local' variables are homeostatically controlled, for example if inhibitory neurons stabilise their own activity (rather than their total excitatory input), or if excitatory-to-inhibitory presynaptic terminals stabilise their average synaptic output (rather than the activity of the excitatory neuron). In these cases, excess activity of excitatory neurons would lead to *decreased* activity of inhibitory neurons, which would be expected to make excitatory neurons *even more* active, thus opposing

homeostasis at the network level. Such 'local' homeostasis of inhibitory activity has been observed in dissociated neurons and brain slices (Bartley et al., 2008; Desai et al., 1999; Gibson et al., 2006; Wenner, 2011). However, it remains unclear whether such local inhibitory homeostasis occurs in the intact central brain *in vivo* and whether it actually causes anti-homeostatic effects on the activity of excitatory neurons.

We address this question in the mushroom body of the fruit fly *Drosophila*. This structure underlies olfactory associative memories, which are stored in the mushroom body's principal neurons, the excitatory Kenyon cells (KCs). KCs respond sparsely to odours (on average ∼5%–10% respond reliably to each odour), and this sparse coding enhances flies' ability to learn to discriminate between similar odours (Campbell et al., 2013; Honegger et al., 2011; Lin et al., 2014; Parnas et al., 2024). The sparseness of KC activity is regulated by the balance of excitation from projection neurons (PNs) and inhibition from a feedback interneuron called the anterior paired lateral neuron (APL) (Lin et al., 2014; Masuda-Nakagawa et al., 2014). This balance is regulated in turn by homeostatic plasticity; when excess inhibition is applied by artificially activating the inhibitory APL, KCs increase their odour responses as compensation (Apostolopoulou & Lin, 2020), through a combination of increased excitation/excitability of KCs and reduced activity of the inhibitory APL. This homeostatic decrease in APL activity is consistent with APL stabilising either its own activity (too high, from artificial activation) or total excitatory synaptic input (too low, from KCs being silenced). However, if APL is activated by artificially

activating not APL itself, but rather KCs, would APL's activity decrease (local homeostasis, e.g. APL stabilising its own activity) or increase (network homeostasis, e.g. APL stabilising the activity of KCs)?

Here we find that when KCs are artificially activated, APL decreases its sensitivity to KC activity. This decrease in turn reduces inhibition onto KCs, which counteracts KCs' own attempts to decrease their activity by reducing excitation. The net result of these conflicting adaptations is that the excess activity does not consistently result in a compensatory decrease in KC odour responses; that is, local adaptation conflicts with network adaptation.

## Methods

### Flies

Flies (*Drosophila melanogaster*) were raised in standard plastic/glass fly vials/bottles, with *ad libitum* access to standard cornmeal agar (80 g medium cornmeal, 18 g dried yeast, 10 g soya flour, 80 g malt extract, 40 g molasses, 8 g agar, 25 ml 10% nipagin in ethanol, 4 ml propionic acid per 1 l water) at 25°C (for preparatory fly crosses) or at 22°C (dTRPA1 flies). Flies were anaesthetised by carbon dioxide via a Flypad (Flystuff) for genetic crosses or on ice before imaging experiments. Either carbon dioxide or ice was used before dissecting the brain out for structural imaging. Flies were collected 0–1 days after eclosion and placed individually in 5 mm diameter glass tubes at 22 or 31°C before imaging/dissection, for the durations described in the Results. Details of fly strains used are given in Tables A1 and A2.

### Functional imaging

Calcium imaging was performed as described (Apostolopoulou & Lin, 2020). Cuticle and trachea in a window overlying the mushroom body were removed, and the exposed brain was superfused (perfusion pump Watson-Marlow [Falmouth, UK] 120S DM2, ~2.7 ml/min) with carbogenated (95% $O_2$, 5% $CO_2$) solution containing 103 mM NaCl, 3 mM KCl, 5 mM trehalose, 10 mM glucose, 26 mM $NaHCO_3$, 1 mM $NaH_2PO_4$, 1.5 mM $CaCl_2$, 4 mM $MgCl_2$, 5 mM TES, pH 7.3. The experimenter was blind to which APL neurons were labelled before postexperimental dissection (Fig. 5), but not otherwise. Odours ($10^{-2}$ or $10^{-4}$ for isoamyl acetate (Sigma [St Louis, MO, USA], 112674), $10^{-2}$ for δ-decalactone (Sigma, W236101); $10^{-1}$ for ethyl butyrate (Fisher [Waltham, MA, USA], 10429220), 4-methylcyclohexanol (Sigma, 153095), 3-octanol (Sigma, 218405), $5*10^{-2}$ for apple cider vinegar (Tesco [Welwyn Garden City, UK]); all odours diluted in mineral oil except for vinegar, which was

diluted in water) were delivered by switching mass-flow controlled carrier and stimulus streams (Sensirion [Stäfa, Switzerland]) via software controlled solenoid valves (The Lee Company [Westbrook, CT, USA]). The flow rate at the fly was ~0.5 l/min. Flies were heated during imaging using a perfusion heater (Scientifica [Uckfield, UK], SM4600). Histamine (2 mM, Sigma H7250) was added 5 min before imaging in APL > Ort experiments as in Apostolopoulou & Lin (2020). To check the stochastic labelling of APL in the APL > Ort experiments, we included UAS-mCherry in the genotype. mCherry expression in APL was distinguished from 3XP3-driven dsRed from the GH146-FLP transgene using separate filter cubes for dsRed (49004, Chroma [Bellows Falls, VT, USA]: 545/25 excitation; 565 dichroic; 605/70 emission) and mCherry (LED-mCherry-A-000, Semrock [Rochester, NY, USA]: 578/21 excitation; 596 dichroic; 641/75 emission).

Brains were imaged using two-photon microscopy (Ng et al., 2002; Wang et al., 2003). Fluorescence was excited using 75–100 fs pulses of 910 nm (for imaging GCaMP6f alone) or 1000 nm (for imaging GCaMP6f and jRGECO1a together) light at 80 MHz from a Ti:Sapphire laser (Spectra-Physics [Milpitas, CA, USA] eHP DS), attenuated by a Pockels cell (Conoptics [Danbury, CT, USA], Model 350–80LA) and coupled to a galvo-resonant scanner on a Movable Objective Microscope (Sutter Instrument, Novato, CA, USA). Excitation light was focused through a 20×, 1.0 NA objective (Olympus [Tokyo, Japan] XLUMPLFLN20XW), and emitted photons were passed through a 750 nm short-pass filter (to exclude excitation light) and band-pass filters (green: 525/50; red: 605/70), and detected by GaAsP photomultiplier tubes (Hamamatsu Photonics [Hamamatsu City, Japan], H10770PA-40SEL), whose currents were amplified (Thorlabs [Newton, NJ, USA], TIA60) and transferred to the imaging computer running ScanImage 5 (Vidrio Technologies [Ashburn, VA, USA]). Volume imaging for Fig. 5 was performed with a piezo objective stage (nPFocus400, nPoint [Middleton, WI, USA]) using ScanImage's FastZ control in sawtooth mode (8 z slices, volume rate ~5.3 Hz). For dual colour imaging, a single z slice was sampled at ~60 Hz and averaged over 10 frames.

Movies were motion-corrected in X–Y using the moco ImageJ plugin (Dubbs et al., 2016), with preprocessing to collapse volume movies in Z and to smooth the image with a Gaussian filter (standard deviation = 4 pixels; the displacements generated from the smoothed movie were then applied to the original, unsmoothed movie), and motion-corrected in Z by maximising the pixel-by-pixel correlation between each volume and the average volume across time points (Bielopolski et al., 2019). $\Delta F/F$ traces were calculated in ImageJ using manually drawn regions of interest (ROIs) for the background and brain structure

of interest, and smoothed with a 0.2 s boxcar filter in Igor Pro 9 (WaveMetrics [Portland, OR, USA]). Traces of APL odour responses were smoothed using a 1 s boxcar filter. Where traces with different frame times needed to be averaged, traces were linearly interpolated to a frame time of 0.018 s. Flies were excluded if the neurons of interest did not respond to odour, the GCaMP6f signal was too low/noisy or the brain moved too much to correct for motion artifacts. Responses were quantified using the mean $\Delta F/F$ over the 5 s odour pulse (equivalent to the integral), rather than the peak, because APL responses were relatively noisy (APL > GCaMP6f signal was dim because APL is only a single neuron, and the laser was set to 1000 nm to excite jRGECO1a and GCaMP6f simultaneously).

## Structural imaging

Dissected brains were fixed in 4% (wt/vol) paraformaldehyde made by 1:5 dilution of 20% paraformaldehyde (Electron Microscopy Sciences [Hatfield, PA, USA], 15713-S) with phosphate-buffered saline made from tablets (PBS-T, Sigma P4417-100TAB, with 0.3% Triton-X100 added), washed in PBS-T (2 quick washes, then 3 20 min washes), blocked with 5% goat serum (Sigma, G6767) in PBS-T, incubated in primary antibody (chicken anti-GFP, Abcam [Cambridge, UK] ab13970, 1:2000; rabbit anti-dsRed, TaKaRa [Kusatsu, Japan], 632496, 1:500) at 4°C over two to three nights, washed in PBS-T (2 quick washes, then 3 20 min washes), incubated in secondary antibody (goat anti-chicken Alexa 488, 1:1000, ThermoFisher [Waltham, MA, USA] A-32931; goat anti-rabbit Alexa 546, 1:1000 Thermo-Fisher A-11071; DAPI, 4 μg/ml, Merck, D9542), washed in PBS-T (2 quick washes, then three 20 min washes) and mounted in Vectashield (Vector Laboratories [Newark, CA, US], H-1000). Brains were imaged using a Nikon A1 confocal in the Wolfson Light Microscopy Facility at the University of Sheffield.

## CRTC::GFP

Zero to 1-day-old flies were placed individually in 5 mm glass tubes in an incubator (MIR-154, PHCbi [Tokyo, Japan]) at (1) 22°C for 24 h, (2) 22°C for 24 h then 31°C for 15 min, (3) 22°C for 18 h then 31°C for 6 h or (4) 31°C for 24 h. Flies were dissected immediately upon removal from the incubator. Cytoplasmic and nuclear GFP signals were segmented manually because initial attempts at automated analysis were unsuccessful. For each hemisphere, the cytoplasm and nucleus of 20 KCs or the single APL neuron were manually outlined using the mCherry signal as a landmark (outlining was carried out blind to the experimental condition), and the average GFP signal in each ROI was quantified, and the background was subtracted. The nuclear localisation index (NLI) was calculated for each cell as described in Bonheur et al. (2023): (nuclear GFP – cytoplasmic GFP)/(nuclear GFP + cytoplasmic GFP). For KCs the NLI for each hemisphere is the average NLI across 20 cells. We treated each hemisphere as a separate sample because we did not observe significant correlations in NLI between the left and right hemispheres of each brain within each experimental condition.

## Para-FlpTag analysis

Mushroom bodies expressing Para-FlpTag were analysed as described (Amin et al., 2020; Greenin-Whitehead et al., 2025) with minor modifications.

**Segmentation and skeletonisation.** A three-dimensional (3-D) volume mask of the mushroom body was created by manually tracing the dsRed signal in every third z slice of the confocal stack, then linearly interpolating from these outlines to all z slices. This volume mask was converted to a 1-D 'backbone skeleton' passing through the centre of the major branches of the mushroom body (vertical lobe, horizontal lobe and peduncle/calyx) by manually defining key anatomical landmarks (e.g. tip of the vertical/horizontal lobes, junction and places where the structure curves) and connecting them into an undirected graph. Tracing and skeletonisation were carried out blind to the experimental condition.

**Spatial standardisation.** To compare the spatial distribution of fluorescence across samples with slightly varying mushroom body sizes, a 'standard' mushroom body backbone was generated by measuring the average length of each of four 'sections': the calyx, peduncle, horizontal lobe and vertical lobe. The boundary between the peduncle and two lobes was defined as the only skeleton node with three edges. The boundary between the calyx and the peduncle was manually defined by drawing a cutting plane where the wide oval shape of the calyx narrowed to the cylindrical shape of the peduncle. Each individual backbone was normalised to this standard backbone by placing nodes at spacing $x*20$ μm, where $x$ is the ratio of the branch length of the individual backbone over the average branch length.

**Volumetric partitioning and quantification.** These evenly spaced backbone nodes were used to define the centroids of Voronoi cells that partitioned the 3-D volume of each mushroom body. This allowed fluorescence to be quantified at regularly spaced intervals along the skeleton. In each z-plane, the background signal in a nearby area of the brain outside the mushroom body

in the same z-plane was subtracted. For each Voronoi cell, background-subtracted GFP and dsRed signals were averaged over all voxels, and the average GFP signal was divided by the dsRed signal to give the normalised Para-FlpTag signal. Because images were taken during two periods separated by several months, we normalised these values to the normalised Para-FlpTag signal averaged across the control flies in the same batch of images. Data from the vertical lobe are not presented due to a large number of outlier segments with negative or extremely high GFP/dsRed ratios, likely resulting from artifacts in how the background fluorescence was measured. We excluded two outliers from the dTRPA1 condition where the average background-subtracted GFP fluorescence was negative, and one outlier from the control condition where the background-subtracted GFP fluorescence was very high; including these outliers would only make our observed effect stronger.

### Computational modelling

**Model generation and parameter tuning.** The computational model of the simplified mushroom body circuit followed that of Abdelrahman et al. (2021). Briefly, we constructed a firing-rate model of 2000 KCs that receive excitation from 24 olfactory PNs and feedback inhibition from a single APL neuron. The inhibitory feedback (activity of the APL neuron) results from the combined input to all KCs independent of KC firing rates, mimicking dendro-dendritic interactions. Thus the firing rate response $y$ of KC $j$ to stimulus $k$ is given by:

$$y_j^k = \text{Relu} \left( \sum_{i=1}^{24} w_{ij} x_i^k - \alpha \sum_{j=1}^{M} \sum_{i=1}^{24} w_{ij} x_i^k - C_\theta \theta_j \right)$$

where $x_i^k$ is the activity of PN $i$ to stimulus $k$, $w_{ij}$ is the weight of the synapse from PN $i$ to KC $j$, $M = 2000$ is the number of KCs, $\theta_j$ is the threshold of KC $j$, Relu is the rectified linear unit, where $\text{Relu}(x) = 0$ if $x \leq 0$ or $x$ if $x > 0$, and the tuneable parameters $\alpha$ and $C_\theta$ adjust the strength of inhibition and the spiking thresholds, respectively (see below). Note that the strength of inhibition $\alpha$ reflects both strength of inputs to APL (which would affect APL calcium responses) and strength of outputs from APL (which would not affect APL calcium responses). To reflect the sparse connectivity from PNs to KCs, each KC received input from a random number of distinct PNs, drawn from a normal distribution with $\mu = 6$, $\sigma = 1.7$ (rounded and truncated to the range [2,11], reflecting results from Caron et al. (2013)). PN-KC synaptic weights were drawn from a log-normal distribution with $\mu = -0.0507$, $\sigma = 0.3527$. The spike thresholds of individual KCs, $\theta_j$, were drawn from a normal distribution with coefficient of variation 0.26, based on Turner et al. (2008).

The excitatory drive from PNs, $x^k$, was derived by propagating the responses of olfactory receptor neurons (ORNs) to a set of 110 odorants (measured by Hallem & Carlson (2011)) to the PNs according to the transfer function proposed by Olsen et al. (2010):

$$x_i^k = R_{\max} \frac{(\text{ORN}_i^k)^{1.5}}{(\text{ORN}_i^k)^{1.5} + (s^k)^{1.5} + \sigma^{1.5}}$$

where $s^k = m \, \Sigma_i \text{ORN}_i^k / 190$, $m = 10.63$ represents the gain of lateral inhibition in the antennal lobe, $R_{\max} = 165$ sets the maximum PN response and $\sigma = 12$ determines the non-linearity of the ORN-PN response function. To generate PN input data for parameter tuning, we added Gaussian-distributed noise to PN activity, scaled by a firing rate-dependent coefficient of variation that was experimentally determined (Bhandawat et al., 2007). Thirty noisy trials were generated for each original extrapolated odour response.

The free parameters $\alpha$ and $C_\theta$ were tuned by imposing the constraints that on average 10% of KCs are active during odour presentation, which rises to 20% upon blocking inhibition, based on Lin et al. (2014). For consistency with our previous work, the excitatory weights $w_{ij}$ were tuned to equalise average activity across all KCs (for details and derivation see Abdelrahman et al. (2021)).

**Modelling plasticity.** Analogous to calcium-imaging experiments, the main readout was the sum of all KC activity to trials of isoamyl acetate. After the initial parameter tuning during model generation, KC properties such as spike threshold $\theta$, excitatory strength $w$ or inhibitory strength $\alpha$ were modified proportionally for the entire KC population, not on the level of individual KCs. To produce a state-space of KC activity $w$ and $\alpha$ were scaled in the range [0, 1.2] and $\theta$ in the range [0.8, 2] in a grid search (where 1.0 keeps the parameter the same as in the base model).

### Statistics

Statistical analyses were performed using Prism 10 (GraphPad [San Diego, CA, USA]) and MATLAB (MathWorks [Natick, MA, USA]). Traces of manual ROIs were analysed in Igor Pro 9 (WaveMetrics). Parametric ($t$ test, ANOVA) or non-parametric tests (Mann–Whitney, Kruskal–Wallis) were used depending on whether residuals passed the D'Agostino-Pearson normality test. For ANOVAs and unpaired $t$ tests, Welch corrections were applied when variances were significantly different between groups. Significant ANOVAs and Kruskal–Wallis tests were followed by *post hoc* tests with multiple comparisons corrections to test the significance of pairwise differences. Random

assignment to experimental groups was not used as all manipulations were genetic. In general, no statistical tests were done to predetermine sample size.

For Fig. 2*D*, the true distribution of KC *vs*. APL correlations was compared to a synthetic distribution created as follows: APL responses were simulated as the true KC data (7 odour responses each for 220 samples) plus Gaussian noise (SD = 0.135). This simulated noise was determined by gradient descent: the SD for each sample was taken from a log-normal distribution, that is $\exp(N(\mu, \sigma))$, where $N$ is a normal distribution with mean $\mu$ and SD $\sigma$. On each iteration if the simulated correlations were higher (lower) than the real correlations, $\mu$ was increased (decreased), and if the variance in simulated correlations was higher (lower) than the variance in the real correlations, $\sigma$ was decreased (increased). This procedure converged on $\mu = -2.0$ and $\sigma = 0$.

To evaluate whether the real data diverged from the simulation more than expected by chance, the simulation was run 1000 times. We calculated the Kolmogorov–Smirnov (K-S) test statistic $D_n$ for every pair of simulation *vs*. simulation and simulation *vs*. data ($D_n$ is the largest absolute difference between the cumulative distribution functions of two datasets). For each simulation, we took the mean $D_n$ between it and the other 999 simulations, and we compared this distribution of 1000 mean simulation-*vs*.-simulation $D_n$ values to the mean simulation-*vs*.-data $D_n$ value. We took the *P*-value to be the fraction of simulation-*vs*.-simulation $D_n$ values that were larger than the simulation-*vs*.-data $D_n$ value. That is, if <5% of the mean simulation-*vs*.-simulation $D_n$ values were larger (more divergent) than the mean model-*vs*.-data $D_n$ value, it would mean the simulation had only a <5% chance of producing results as divergent from the simulation as the real data and thus the simulation and data could be said to be significantly different ($P < 0.05$). This method is essentially a Monte Carlo version of a two-sample K-S test.

## Results

### Decreased APL sensitivity following excess Kenyon cell activity

To measure APL's sensitivity to KC input, we simultaneously recorded from KCs and APL. We used the binary expression systems GAL4/UAS and LexA/lexAop (Venken et al., 2011) to express the green calcium indicator GCaMP6f (Chen et al., 2013) in APL using the driver VT43924-GAL4.2 (Amin et al., 2020), and to express the red calcium indicator jRGECO1a (Dana et al., 2016) in KCs using the driver mb247-LexA (Pitman et al., 2011). We artificially activated KCs using the heat-activated cation channel dTRPA1, which is normally expressed in *Drosophila* heat-sensing neurons (Hamada

et al., 2008). Ectopically expressing dTRPA1 in KCs allowed us to control their activity by raising the ambient temperature. This thermogenetic tool produces more sustained neuronal activation than channelrhodopsin (Pulver et al., 2009) and has previously been used to induce homeostatic plasticity (Apostolopoulou & Lin, 2020; Oswald et al., 2018). In agreement with previous results (Amin et al., 2020; Apostolopoulou & Lin, 2020; Lin et al., 2014), raising the temperature to ∼31°C significantly increased calcium indicator signal in both KCs (direct activation) and APL (excited by KCs), as measured in the calyx, where KC dendrites reside (Fig. 1*A* and *B*). We showed previously that in flies without dTRPA1 in KCs, raising the temperature during the recording does not increase baseline APL $Ca^{2+}$ influx (Amin et al., 2020; Lin et al., 2014), confirming that the effect is due to opening dTRPA1 rather than heating alone.

Although it was not feasible to continuously image the mushroom body for an extended period, we tested longer-term activation by holding flies at ∼31°C for ∼30 min without imaging to avoid bleaching, then recording the fall in $Ca^{2+}$ during cooling. In the cooling recordings, KC > jRGECO1a and APL > GCaMP6f fluorescence both started high and fell to a stable baseline, suggesting that $Ca^{2+}$ influx remained high even after 30 min of heating. After heating flies a second time, $Ca^{2+}$ signals increased significantly again in both KCs and APL (Fig. 1*A* and *B*). The KC response was slightly higher in the later recordings, whereas the APL response was slightly lower, but this difference was not statistically significant (two-way repeated-measures ANOVA, $P = 0.123$ for interaction between cell type and time). To test whether dTRPA1-driven $Ca^{2+}$ influx is sustained beyond this ∼30 min, we used the activity-dependent transcription factor CRTC (CREB-regulated transcriptional coactivator), which is imported into the nucleus when neurons are active, and exported to the cytoplasm when neurons are inactive, C on the timescale of minutes (Bonheur et al., 2023). We expressed CRTC::GFP in KCs using the driver OK107-GAL4; KC somata could be clearly identified as the small cells clustered around the calyx (Fig. 1*C* and *D*). After 15 min at 31°C, CRTC::GFP in KCs was more localised to the nucleus compared to flies left at 22°C, and the nuclear localisation of CRTC::GFP was sustained after 24 h at 31°C (Fig. 1*E* and *F*), indicating that dTRPA1 induces prolonged activation of KCs. dTRPA1 had no effect in flies left at 22°C. This effect was not due to a selection effect from differential penetrance of CRTC::GFP expression in KCs, as OK107-GAL4 labels all ∼2000 KCs per hemisphere (Aso et al., 2009; Greenin-Whitehead et al., 2025), and ∼2000 KCs were labelled in both control and dTRPA1 flies (Fig. 1*G*).

We measured CRTC::GFP localisation in APL using the driver VT43924-GAL4.2; although some non-APL

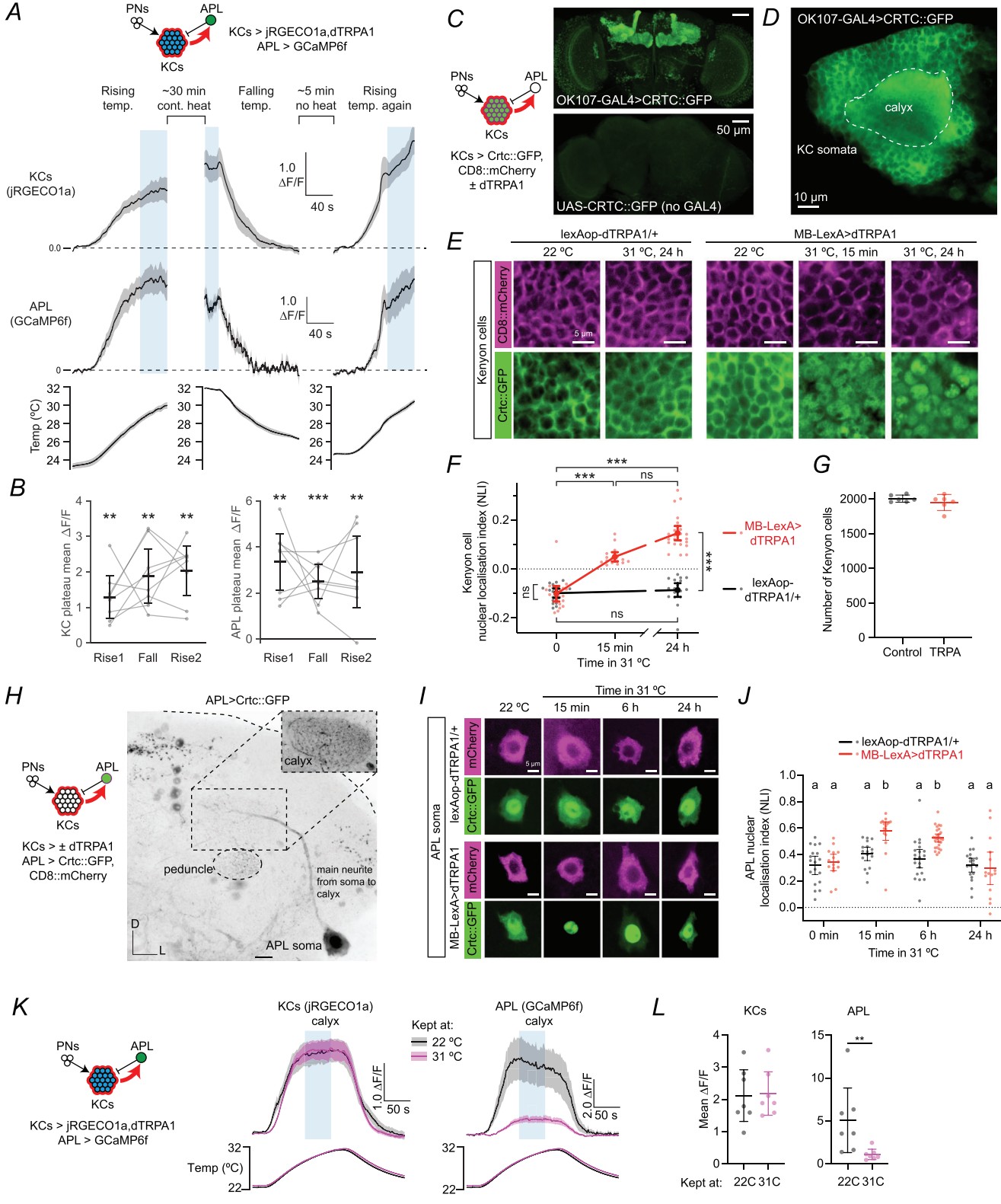

**Figure 1. Decreased anterior paired lateral (APL) sensitivity following excess Kenyon cell (KC) activity**

*A*, response of KCs and APL to heat in flies expressing jRGECO1a and dTRPA1 in KCs, and GCaMP6f in APL, raised at 22°C, imaged in the calyx. Flies were imaged during heating, held at ~31–32°C for 30 min without imaging, imaged during cooling and heated again. ΔF/F was normalised to the baseline fluorescence within each movie. Blue shaded regions indicate time periods for quantification in (*B*). Error shading shows SEM. *N* = 8. *B*, mean ΔF/F

during the shaded periods indicated in (*A*) (rise 1: 100–140 s; fall: 0–20 s; rise 2: 80–120 s). ** *P* < 0.01, *** *P* < 0.001, one-sample *t* test with Holm–Bonferroni multiple comparisons correction. Two-way repeated-measures ANOVA revealed no significant interaction between cell type (KCs *vs.* APL) and time (rise 1 *vs.* fall *vs.* rise 2) (*P* = 0.123). *C*, maximum intensity z-projection of CRTC::GFP expressed in KCs (upper) compared to a negative control brain with UAS-CRTC::GFP (no GAL4 driver) (lower panel), both stained with anti-GFP. Scale bars = 50 μm. *D*, example confocal slice of CRTC::GFP expression in KCs driven by OK107-GAL4. (This fly also expressed dTRPA1 driven by mb247-LexA and was kept at 22°C.) KC somata were identified as the small cells tightly clustered around the calyx (dashed white outline). Cells outside this cluster (e.g. lower right corner) were not quantified. *E*, CRTC::GFP (green) and CD8::mCherry (magenta) driven by OK107-GAL4 in KC somata expressing lexAop-dTRPA1 driven by mb247-LexA. All flies were raised at 22°C. Scale bar = 5 μm. *F*, quantification of CRTC::GFP localisation in KCs expressing dTRPA (red) or not (black). Nuclear localisation index (NLI) was calculated as (mean nuclear GFP signal − mean cytoplasmic GFP signal)/(mean nuclear GFP signal + mean cytoplasmic GFP signal). *** *P* < 0.001, Kruskal–Wallis test and Dunn's multiple comparisons test. *N* expressed as # hemispheres (# flies): 17(9), 17(10) (Ctrl); 19(10), 16(8), 24(12) (dTRPA1). *G*, ∼2000 KCs are labelled by OK107-GAL4 driving CRTC::GFP and CD8::mCherry in both control and dTRPA1 flies kept at 31°C for 24 h (*n* = 6,6; *P* = 0.667, Mann–Whitney test). *H*, example confocal image (inverted colour scale) of CRTC::GFP expression in APL driven by VT43924-GAL4.2; same image as lexAop-dTRPA/+, 15 min at 31°C in (*I*). Maximum intensity z-projection over 63 μm (21 slices) (does not include the mushroom body lobes). Inset shows a single confocal slice of the calyx, which is not easily visible in the z-projection. Dashed oval outlines the cross-section of the peduncle. Scale bar = 10 μm. D, dorsal; L, lateral. *I*, *J*, as (*F*, *G*) but with CRTC, mCherry in APL driven by VT43924-GAL4.2. *N*: 18(10), 15(9), 17(9), 17(9), 21(12), 27(14), 19(10), 15(8). a, b, groups with different letters are significantly different; groups with the same letters are not. *K*, response of KCs and APL to heat in KC > jRGECO1a,dTRPA; APL > GCaMP6f flies kept at 22°C (black) or 31°C (magenta) for 24 h, imaged in the calyx. Shading shows SEM. *L*, mean Δ*F/F* during the shaded time window (100–150 s) of the responses in (*K*). *N* = 7,7. Graphs in (*B*), (*F*), (*G*), (*J*) and (*L*) show mean ± 95% CI. See Table A3 for detailed statistics and exact *P*-values.

neurons were labelled, APL could be clearly recognised as a large cell body on the lateral edge of the midbrain, with a thick neurite extending towards the calyx, which was filled with APL's characteristic reticulated neurites (Fig. 1*H*) (compare to Lin et al. (2014); Liu & Davis (2008)). Both APL neurons (i.e. one per hemisphere) were labelled in all brains examined. APL's CRTC::GFP was more localised to the nucleus after 15 min and 6 h at 31°C, but the nuclear localisation returned to baseline levels after 24 h (Fig. 1*I* and *J*), suggesting that APL is less active after 24 h stimulation despite continued activity in KCs.

Because calcium influx in APL is localised (Amin et al., 2020), CRTC nuclear localisation in the APL soma may not fully report calcium influx elsewhere in APL. Therefore, we measured APL and KC activity in the calyx during dTRPA1 activation of KCs in flies preheated to 31°C for 24 h, compared to flies kept at 22°C. Although the response to heat was the same in KCs in flies kept at 22 *vs.* 31°C, the response in APL was smaller (but not zero) in preheated flies (Fig. 1*K* and *L*). Together, our CRTC and calcium imaging results suggest that APL is less sensitive to excitation from KCs after prolonged KC activity.

Next we turned to more naturalistic odour stimuli; we measured simultaneous KC and APL responses to a panel of seven odours of differing intensities (Fig. 2*A* and *B*), quantified the responses as the mean Δ*F/F* during the 5 s odour stimulus and plotted the responses against each other in a scatter plot (Fig. 2*C*). We recorded odour responses in the different anatomical regions of the mushroom body: the calyx (KC dendrites) and the axonal lobes, formed of the axons of the three main subtypes of KCs, $\alpha\beta$ (forming the $\alpha$ and $\beta$ lobes), $\alpha'\beta'$

($\alpha'$ and $\beta'$ lobes) and $\gamma$ (just the $\gamma$ lobe); the three subtypes differ in their electrical properties (Groschner et al., 2018; Inada et al., 2017; Turner et al., 2008) and their role in different phases of memory (Guven-Ozkan & Davis, 2014). In general, the KC responses and APL responses within one sample were strongly positively correlated, except in the $\gamma$ lobe in preheated KC>dTRPA1 flies, where both KC and APL responses were close to zero (see Figs 3, A1 and A2). Indeed, the distribution of KC *vs.* APL correlation values closely matched the distribution of correlations resulting from simulating APL responses as being simply KC responses plus Gaussian noise (Fig. 2*D*).

Inspecting all scatter plots, we did not observe consistent deviations from linearity, so we chose the most parsimonious model to fit the scatter plots: a linear relationship, or APL = *k*\*KC, where *k* is the slope of the best-fit line through the seven points for each fly (Fig. 2*C*), constrained to go through the origin to match the finding that APL does not respond to odours when KCs are silenced (Lin et al., 2014). We interpret the slope of this line as reflecting the APL's sensitivity to KC activity, because KCs are the primary source of synaptic excitation to APL: KC>APL synapses outnumber PN > APL synapses by 9.5:1 in the calyx and 36:1 overall (Amin et al., 2020; Scheffer et al., 2020). Indeed, in the preheated condition (but not the non-preheated condition), the slope was shallower in KC > dTRPA1 flies than control flies in every case (the difference was statistically significant in the calyx and the $\alpha'$, $\beta'$ and $\gamma$ lobes; Fig. 2*E*–*J*), indicating that after overactivation of KCs APL became less sensitive to KC input.

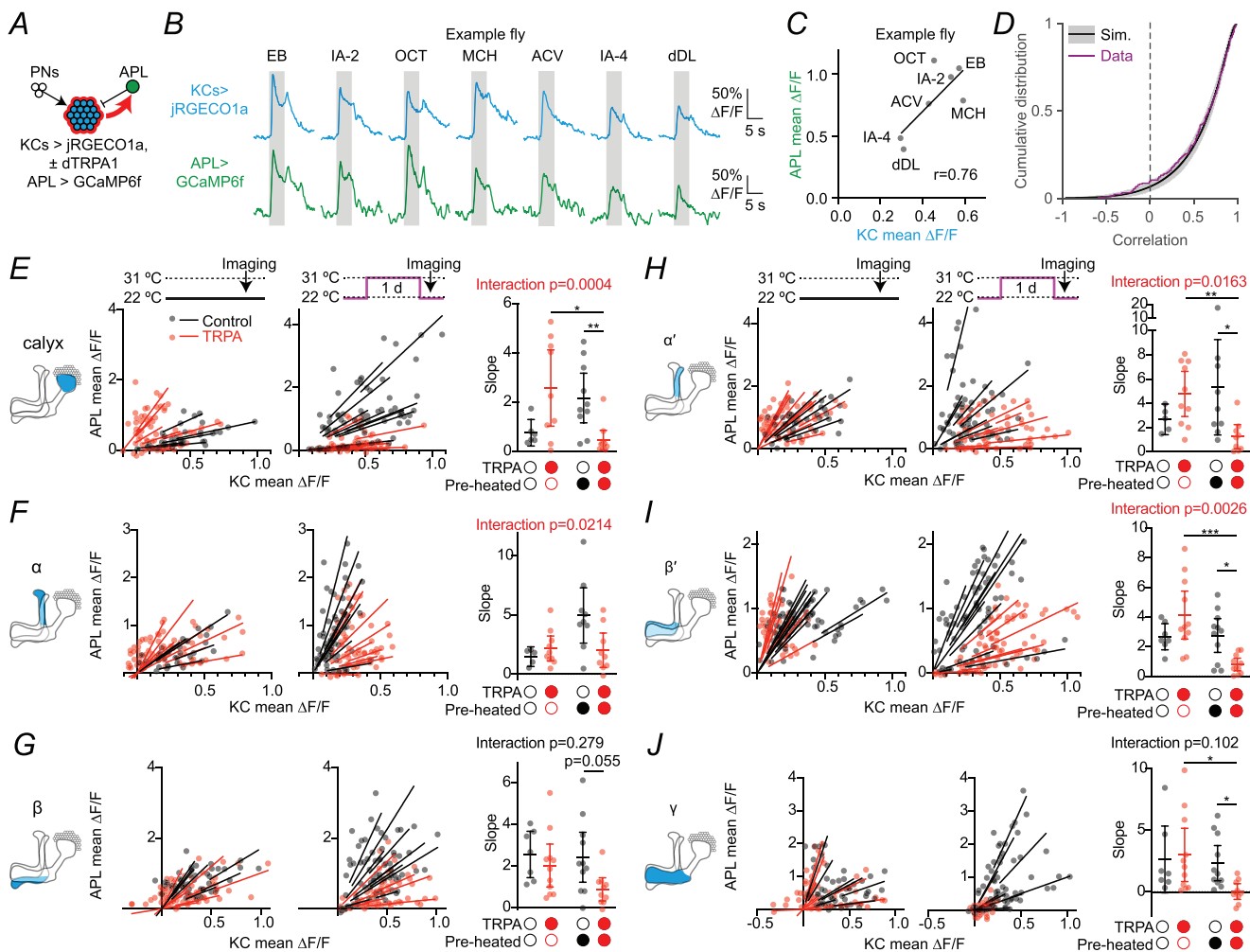

**Figure 2. Decreased anterior paired lateral (APL) sensitivity in odour responses following excess Kenyon cell (KC) activity**

*A*, schematic of experiment. *B*, responses in KCs (blue) and APL (green) to ethyl butyrate $10^{-1}$ (EB), isoamyl acetate $10^{-2}$ (IA-2), 3-octanol $10^{-1}$ (OCT), 4-methylcyclohexanol $10^{-1}$ (MCH), apple cider vinegar $5*10^{-2}$ (ACV), isoamyl acetate $10^{-4}$ (IA-4) and delta-decalactone $10^{-2}$ (dDL), recorded in the calyx of a single example fly without dTRPA1, kept at 22°C. Shaded regions indicate timing of odour stimulus. *C*, scatter plot of responses from (B), quantified as the mean $\Delta F/F$ during the odour (shaded regions in (B)). The best-fit line is a linear regression constrained to go through the origin. *D*, cumulative distribution of correlations between APL *vs.* KC responses, from data (blue) or from a simulation (black) where APL responses equal KC responses plus Gaussian noise (standard deviation 0.135) (see Methods for details). Each sample is one lobe from one fly ($n = 222$). Note this includes $\gamma$ lobe recordings from preheated KC > dTRPA1 flies, where the correlations were near zero because the responses were so low. Distributions binned at 0.01 intervals. The black line is the mean of 1000 simulations (each one simulating 220 samples); 95% of simulations lie within the grey shading. The data do not differ significantly from the simulations ($P = 0.71$, see Methods for test details). *E–J*, scatter plots and slopes of best-fit lines for the calyx (*E*) and the $\alpha$ (*F*), $\beta$ (*G*), $\alpha'$ (*H*), $\beta'$ (*I*) and $\gamma$ (*J*) lobes for flies with (red) or without (black) dTRPA1 in KCs. Scatter plots show the mean $\Delta F/F$ of the odour response in APL *vs.* KCs for flies kept at 22°C (left panels) or 31°C (middle panels) for 24 h before imaging. Each best-fit line is the linear regression for the seven odour responses for one fly. The slopes of the best-fit lines are shown in the right panels. The traces of all odour responses are shown in Figs A1 and A2. *N* expressed as # hemispheres (# flies): 6 (6), 9 (8), 10 (10), 11 (11) (calyx); 5 (5), 10 (9), 9 (9), 9 (9) ($\alpha$, $\alpha'$); 7 (6), 11 (1), 11 (11), 10 (10) ($\beta$, $\gamma$); 7 (6),11 (10),11 (11),11 (11) ($\beta'$). * $P < 0.05$, ** $P < 0.01$, *** $P < 0.001$ (see Table A3 for detailed statistics and exact *P*-values). Graphs show mean $\pm$ 95% CI.

## Heterogeneous plasticity in Kenyon cell odour responses after excess stimulation

How does KC activity change following sustained dTRPA1 stimulation? Given that APL odour responses are almost completely abolished in the calyx where KC dendrites reside (Fig. 3*A* and *C*; two-way ANOVA, interaction between genotype and preheating, $P = 0.006$), one might expect KC odour responses to be greatly increased given this loss of inhibition. After all, silencing APL increases KC GCaMP responses by two to threefold (Apostolopoulou & Lin, 2020; Lin et al., 2014). On the other hand, given that KCs compensate for excess inhibition by becoming more active (Apostolopoulou & Lin, 2020), one might predict that they would compensate for excess *excitation* by becoming *less* active.

In fact, neither scenario held consistently (Fig. 3*B* and *D*). In $\gamma$ KC axons, odour responses were lower in preheated dTRPA1 flies than preheated controls, but this difference was not significantly different from the difference in non-preheated flies (two-way ANOVA, interaction $P = 0.22$). In the calyx (where KC dendrites reside) and in $\alpha\beta$ and $\alpha'\beta'$ KC axons, odour responses were either unaffected or somewhat increased in the $\alpha$ and $\alpha'$ lobes (though in two-way ANOVAs, the interaction between genotype and preheating was only statistically significant in the $\alpha'$ lobe). The lack of a consistent dramatic increase in KC odour responses was not due to jRGECO1a becoming saturated, because we also did not observe increased KC odour responses with dTRPA1 activation using GCaMP6f, in contrast to drastically increased GCaMP6f responses when APL is silenced (see Fig. 5).

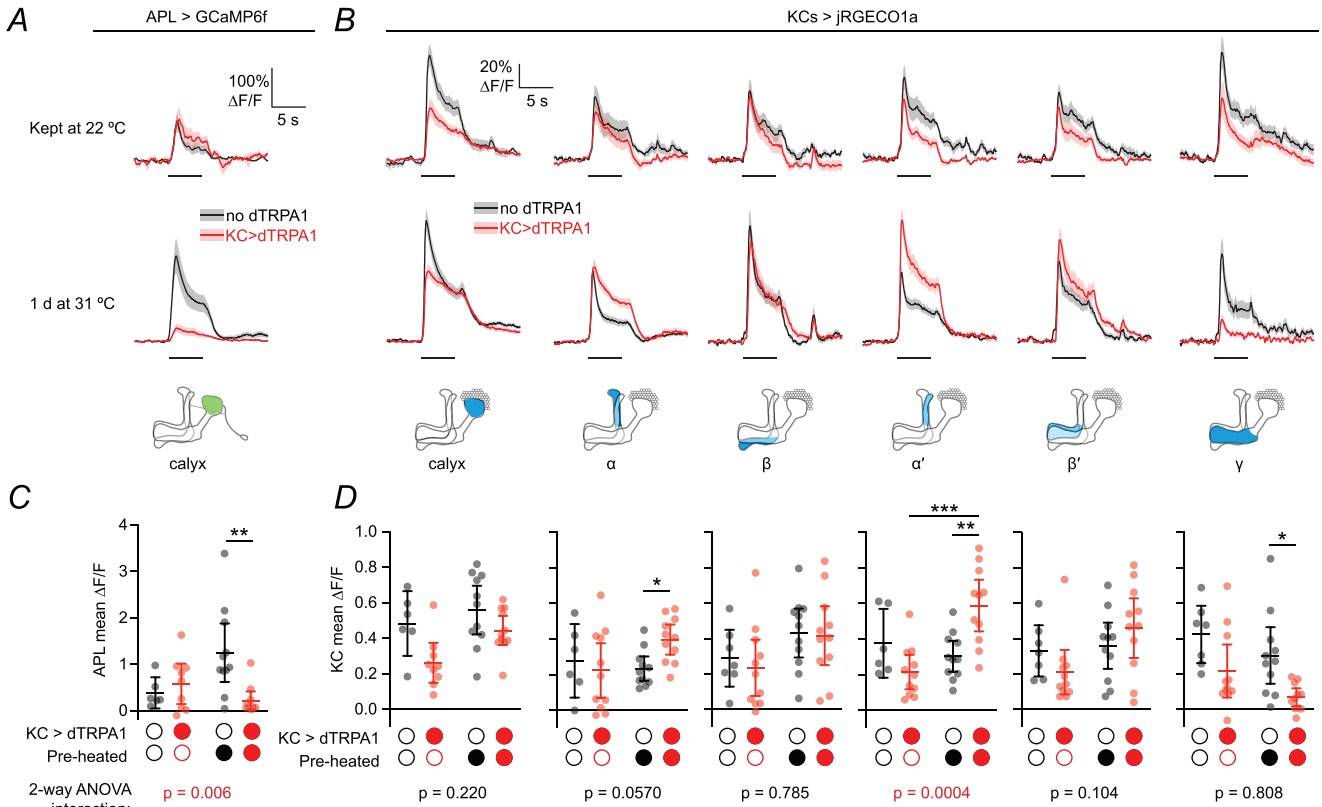

**Figure 3. Kenyon cell (KC) and anterior paired lateral neuron (APL) odour responses after adaptation to Kenyon cell overactivation**

*A*, traces of responses to 5 s $10^{-2}$ isoamyl acetate in the anterior paired lateral neuron (APL) expressing GCaMP6f in the calyx in flies where Kenyon cells express dTRPA1 (red) or not (black), in flies kept at 22°C (top row) or 31°C (middle row) for 24 h before the experiment. Error shading represents SEM; horizontal line indicates the odour presentation. Lobe responses are shown in Fig. A2. *B*, as for (*A*), but responses in KCs expressing jRGECO1a, in different regions of the mushroom body as shown by the diagrams (bottom row). *C*, *D*, mean *ΔF/F* during the 5 s odour presentation from (*A*, *B*). * $P < 0.05$, ** $P < 0.01$, *** $P < 0.001$ (see Table A3 for detailed statistics and exact *P*-values). *N* expressed as # hemispheres (# flies): 6 (6), 9 (8), 11 (11), 11 (11) (calyx); 7 (6), 11 (10), 11 (11), 10 (10) ($\gamma$ APL); 7 (6), 11 (10),11 (11), 11 (11) ($\beta$, $\beta'$, $\gamma$ KCs); 6 (5), 11 (10), 11 (11), 11 (11) ($\alpha$, $\alpha'$). Graphs show mean ± 95% CI.

## Adaptation lowers sodium channel expression in Kenyon cells

Why does decreased APL sensitivity not lead to increased KC activity? Perhaps KCs simultaneously decrease their excitability. As an initial test of this hypothesis, we measured the KC expression of Para, the fly's only voltage-gated Na$^+$ channel (Loughney et al., 1989; Ravenscroft et al., 2020). Para expression increases (decreases) in response to decreased (increased) synaptic excitation (Mee et al., 2004), and ectopic expression of the bacterial sodium channel NaChBac causes decreased Para expression and a concomitant loss of action potentials in KCs (Greenin-Whitehead et al., 2025).

We tagged endogenous Para with GFP specifically in KCs using the FlpTag system (Fendl et al., 2020) (Fig. 4*A*), in which GFP is spliced into Para only in cells expressing FLP recombinase (here KCs expressed FLP driven by mb247-LexA). In this *para*$^{FlpTag}$ allele, the FlpTag cassette

is inserted into a location that labels >98% of *para* transcripts (Ravenscroft et al., 2020), so almost all isoforms of Para should be labelled. This technique reveals both the level and localisation of Para in specific cells, unlike other techniques such as immunolabelling, *in situ* hybridisation, qPCR, RNA-seq or Western blotting. We selected flies heterozygous for *para*$^{FlpTag}$ to mitigate any loss of function from the GFP insertion (note that *para*$^{FlpTag}$ is homozygous viable).

We normalised GFP signal to pan-mushroom-body dsRed signal (Riemensperger et al., 2005) to quantify the relative abundance of Para. Consistent with previous descriptions of Para localisation and the axon initial segment in KCs (Greenin-Whitehead et al., 2025; Ravenscroft et al., 2020; Trunova et al., 2011) and other neurons (Rey et al., 2023), we observed little GFP fluorescence in the calyx but strong signal in the posterior peduncle (Fig. 4*B* and *D*), where numerous molecular

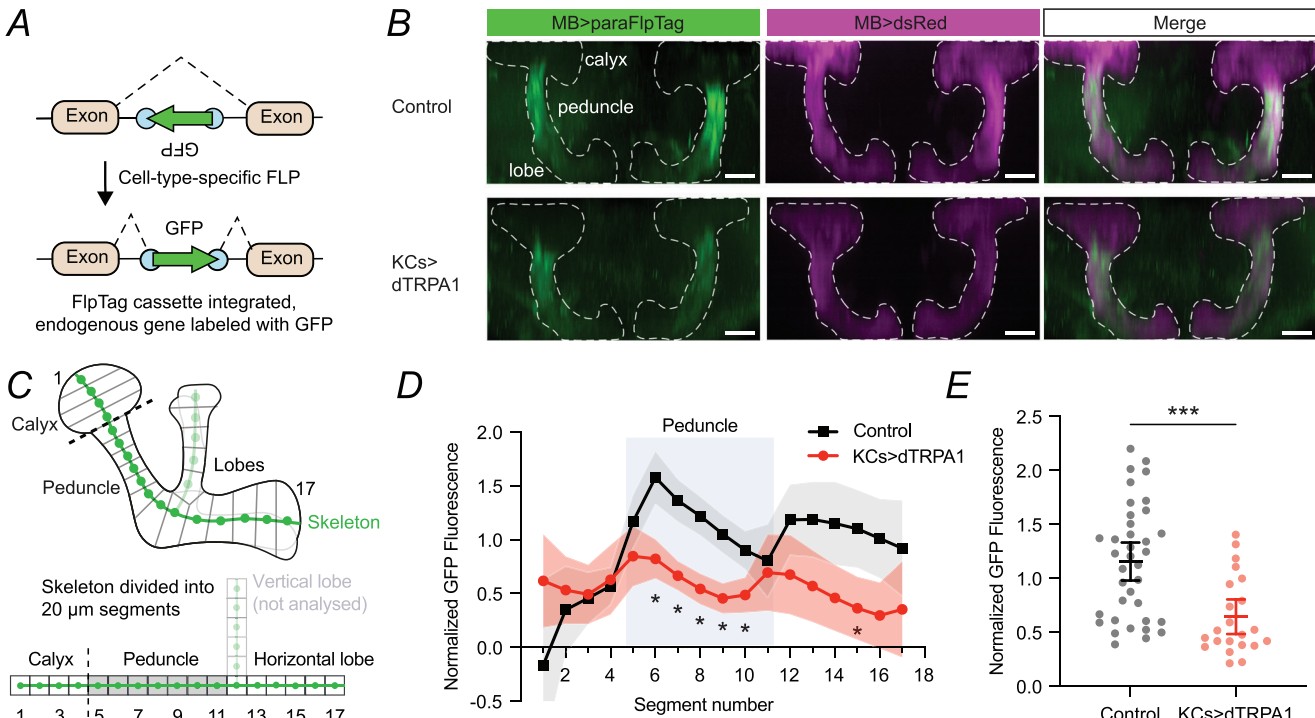

**Figure 4. Kenyon cells (KCs) reduce expression of Para Na$^+$ channels**

*A*, schematic of cell-specific labelling of endogenous proteins using FlpTag. *B*, example control and Kenyon cell (KC) > dTRPA1 mushroom bodies labelled with Para-FlpTag (green) and dsRed (magenta). Images are maximum-intensity projections through the entire mushroom body (confocal stack resliced to be viewed dorsally). Dashed white lines are outlines of the mushroom bodies. Scale bar = 25 μm. *C*, schematic of how the mushroom body skeleton was divided into evenly spaced segments to quantify Para localisation in a diagram of the mushroom body (upper) and more schematic form (lower). The black dashed line is the boundary between the calyx and the peduncle. The green line is the backbone skeleton. The vertical lobe was not analysed (see Methods). *D*, ratio of average GFP signal to dsRed signal in each segment, normalised to the average ratio across segments for control flies. All flies were heated to 31°C for 24 h before dissection. Para expression was lower in KC > dTRPA1 flies, particularly in the peduncle (shaded zone). * $P < 0.05$, two-way repeated-measures ANOVA with Sidak's multiple comparisons test. *E*, mean normalised GFP signal in the peduncle in control (black) *vs*. KC > dTRPA1 (red) flies. *** $P < 0.001$, unpaired *t* test. *N* expressed as # hemispheres (# flies): 37 (19) (Ctrl); 22 (11) (dTRPA1). Graphs show mean ± 95% CI. See Table A3 for detailed statistics and exact *P*-values.

markers indicate that the axon initial segment is located (Trunova et al., 2011). The Para-FlpTag signal was strongly reduced in flies where KCs had been overactivated by dTRPA1 for 24 h, compared to no-dTRPA1 controls. This effect was especially prominent in the peduncle (Fig. 4*C*–*E*). These results suggest that to compensate for overactivation, KCs decrease expression of voltage-gated sodium channels.

### Reduced inhibition from APL counteracts reduced excitation of Kenyon cells

Despite this loss of Para, KC odour responses did not become smaller after dTRPA1 activation (except in *γ* KCs). Could this be explained by reduced inhibition from APL masking an underlying decrease in KC excitation? If so, then removing inhibition from APL would reveal the effects of excitation on KC activity and might thereby unmask a hidden reduction in KC excitation, by allowing

us to compare the activity of control and adapted KCs with inhibition blocked in each case.

To test this prediction, we acutely blocked APL in adapted flies by expressing the histamine-gated Cl⁻ channel Ort in APL and bath-applying histamine (Fig. 5*A*), a manipulation that acutely suppresses APL activity (Amin et al., 2020; Liu & Wilson, 2013). Because we drove Ort expression using a stochastic intersectional driver (Apostolopoulou & Lin, 2020; Lin et al., 2014) (see Methods), some hemispheres did not express Ort in APL; these served as no-Ort negative controls.

We made three main observations. First, in APL > Ort flies (but not in no-Ort negative controls; Fig. A3), histamine drastically increased KC odour responses (Fig. 5*B* and *C*), consistent with our previous findings with silencing APL using Ort, tetanus toxin or shibire (Amin et al., 2020; Apostolopoulou & Lin, 2020; Lin et al., 2014). Second, histamine had a smaller (or indeed, no) effect on KC odour responses in KC > dTRPA1 flies compared to

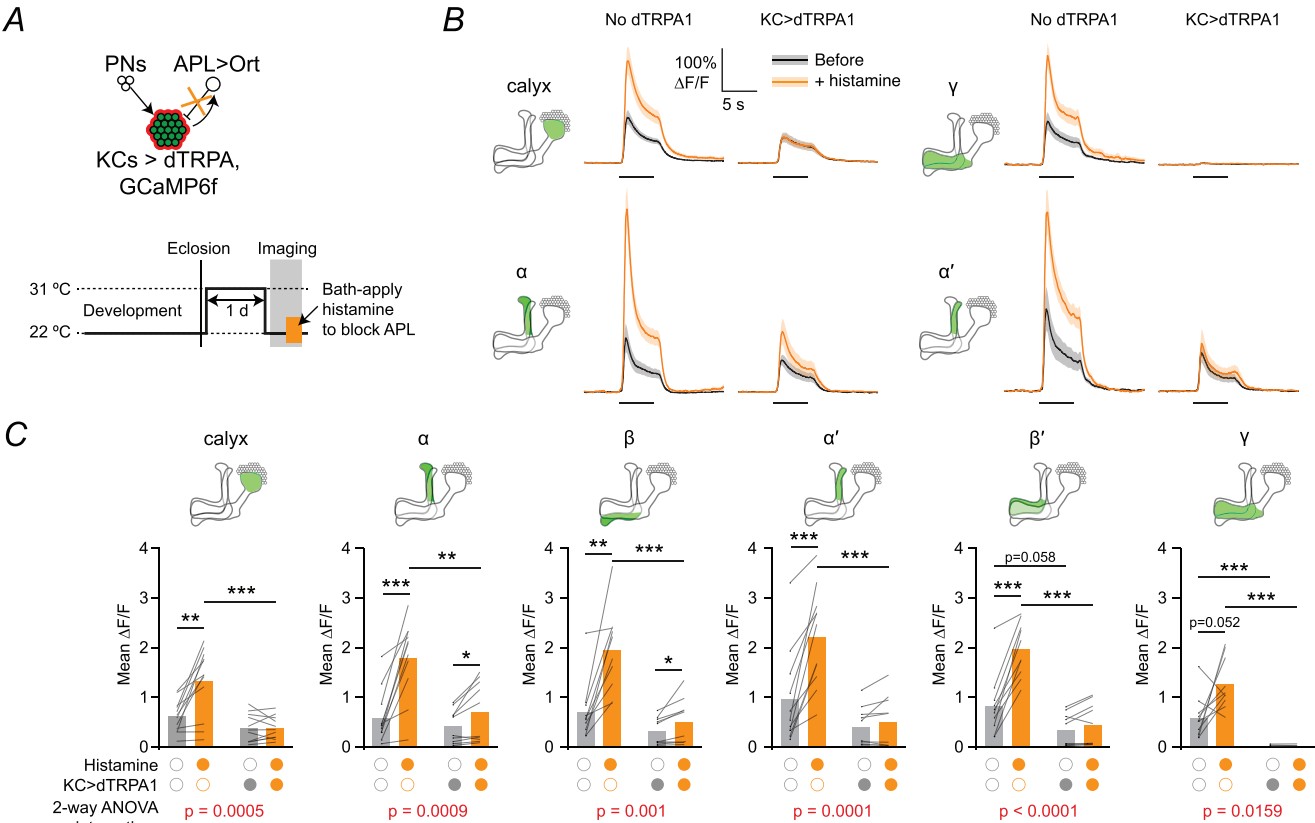

**Figure 5. Removing inhibition from anterior paired lateral (APL) reveals hidden loss of excitation**
*A*, schematic of the experiment. Kenyon cells (KCs) expressed GCaMP6f and dTRPA1, whereas APL expressed Ort. Flies were imaged at 22°C after 24 h at 31°C, and histamine was bath-applied during the experiment. *B*, traces of responses to 5 s 10⁻² isoamyl acetate in flies where KCs express dTRPA1 (right) or not (left) before (black) or after (orange) histamine was applied. Shown here are responses in the calyx and the *α*, *α'* and *γ* lobes; see Fig. A3 for all responses, including control flies without Ort in APL. *C*, mean Δ*F/F* during the 5 s odour presentation from (*B*). *N* expressed as # hemispheres (# flies): *N* = 12 (11), 11 (10) (calyx); 11 (10), 10 (9) (*α*, *α'*); 10 (9),10 (9) (*β*, *β'*, *γ*). Graphs show means, with lines showing individual hemispheres. * *P* < 0.05, ** *P* < 0.01, *** *P* < 0.001 (see Table A3 for detailed statistics and exact *P*-values).

no-dTRPA1 controls (Fig. 5*B* and *C*), implying that APL was providing less (or no) inhibition in the no-histamine condition, consistent with the decreased sensitivity of APL shown in Figs 1–2.

Third, as we hypothesised, blocking APL indeed unmasked reduced excitation onto KCs: when APL was blocked by histamine, KC activity was drastically lower in KC > dTRPA1 flies than no-dTRPA1 controls (compare the orange traces under 'No dTRPA1' *vs.* orange traces under 'KC>dTRPA1', Fig. 5*B*, or compare the orange bars in Fig. 5*C*). This result indicates that in the absence of inhibition, KCs that have been previously overactivated by dTRPA1 receive less synaptic excitation and/or are less intrinsically excitable than control KCs. This change is consistent with the loss of Para (Fig. 4), but importantly, this means that two forms of plasticity counteract each other: KCs reduce their excitability, and simultaneously APL also reduces its activity, thus reducing inhibition onto KCs and cancelling out KCs' reduced excitability.

### Model network reproduces heterogeneous net effects of conflicting adaptations

The net adaptation was heterogeneous between KC subtypes (Figs 3 and 5), consistent with differences in adaptation between neuronal types previously observed in the mushroom body (Apostolopoulou & Lin, 2020) and other systems (Bartley et al., 2008; Galliano et al., 2021; Wen & Turrigiano, 2024; Xue et al., 2014). We also observed somewhat different results in different experiments; for example, compare how KC > dTRPA1 adaptation affected KC odour responses in Figs 3 and 5. These intersubtype and interexperiment differences might arise from slight differences in the balance between the two opposing adaptations (reduced excitation and reduced inhibition), much as how slight changes in the detailed balance of synaptic excitation and inhibition can gate or un-gate neural signals (Vogels & Abbott, 2009).

To test the plausibility of this interpretation, we simulated plasticity in network parameters using our previously published rate-coding model of the mushroom body (Abdelrahman et al., 2021) (see Methods). Here, KCs receive excitation from olfactory PNs and dendro-dendritic feedback inhibition from APL (based on findings of localised inhibition by APL (Amin et al., 2020)) (Fig. 6*A*). Synaptic weights and thresholds were tuned to produce biologically realistic KC odour responses (Hallem & Carlson, 2011; Olsen et al., 2010) for 20 model instances with noisy sensory inputs. We modelled decreasing excitation of KCs following KC dTRPA1 activation (Fig. 4) by scaling down the excitatory weights from PNs or increasing KCs' firing thresholds. We modelled decreasing activity in APL (Figs 1 and 2) by scaling down the feedback inhibitory weight, and

we modelled the complete loss of inhibition caused by blocking APL with Ort (Fig. 5) as simply setting inhibition to zero.

We explored the model's parameter space by systematically scaling up/down the excitation and inhibition and simulating the total KC response to iso-amyl acetate. As expected, when excitatory weights were reduced by ∼15%–25%, or when thresholds were raised by 20%–30% (moving 'west' on the heat maps in Fig. 6*B* and *C*), KC activity dropped to zero regardless of the inhibitory strength, as total synaptic excitation fell below the KCs' thresholds. For any given amount of excitation, reducing inhibition (moving 'south' in Fig. 6*B* and *C*) increased KC activity. Starting from the 'normal' parameter settings (square in Fig. 6*B* and *C*), one can reduce both excitation and inhibition by moving 'southwest'. Notably, depending on the angle of this 'plasticity arrow', this movement can result in increased, decreased or unchanged KC activity (Fig. 6*D*). In particular, by scaling excitation differently while scaling inhibition by a constant amount, we could reproduce the experimentally observed heterogeneous dTRPA1-induced changes in the odour responses of $\alpha\beta$, $\alpha'\beta'$ and $\gamma$ KCs (Fig. 6*E*, compare to Fig. 5*C*). The increased responses in the $\alpha$ and $\alpha'$ lobe in Fig. 3*D* could also be reproduced (star marked 'Incr' in Fig. 6*D*). In other words, depending on the balance of reducing excitation and reducing inhibition, the net effect on KCs can be increased, decreased or unchanged activity.

## Discussion

We have shown that the mushroom body compensates for excess activity in KCs through two conflicting homeostatic effects. Excitation of KCs is reduced, but this is cancelled out by reduced feedback inhibition from APL. The net effect is that KC odour responses are not consistently decreased after adaptation, as would be predicted by homeostatic logic. These results suggest that homeostatic plasticity may follow rules that make sense locally (e.g. compensating for excess activation of APL) but are counterproductive more broadly.

### Adaptation mechanisms

The decreased sensitivity of APL (Figs 1 and 2) might occur through presynaptic mechanisms (e.g. reduced vesicle release at KC-APL synapses) and/or postsynaptic mechanisms (e.g. reduced expression of nicotinic acetylcholine receptors or voltage-gated calcium channels). Both types have been described in other systems (Bartley et al., 2008; Doyle et al., 2010; Moulder et al., 2004). Future experiments will clarify which of these mechanisms occurs (perhaps both), but in any case, the reduced

APL activity translates into reduced feedback inhibition, which masks a loss of odour-evoked excitation in KCs, which we exposed by removing inhibition from APL after adaptation (Fig. 5). This loss of excitation may be attributable in part to the reduced expression of Para voltage-gated $Na^+$ channels (Fig. 4); if this translates to a reduced voltage-gated $Na^+$ conductance, that should make it less likely for KCs to fire action potentials (Greenin-Whitehead et al., 2025). Similar compensatory changes in intrinsic excitability have frequently been observed in other systems (Baines et al., 2001; Baines, 2003; Bartley et al., 2008; Desai et al., 1999; Gibson et al., 2006; Grubb & Burrone, 2010; Maffei & Turrigiano, 2008; Mee et al., 2004).

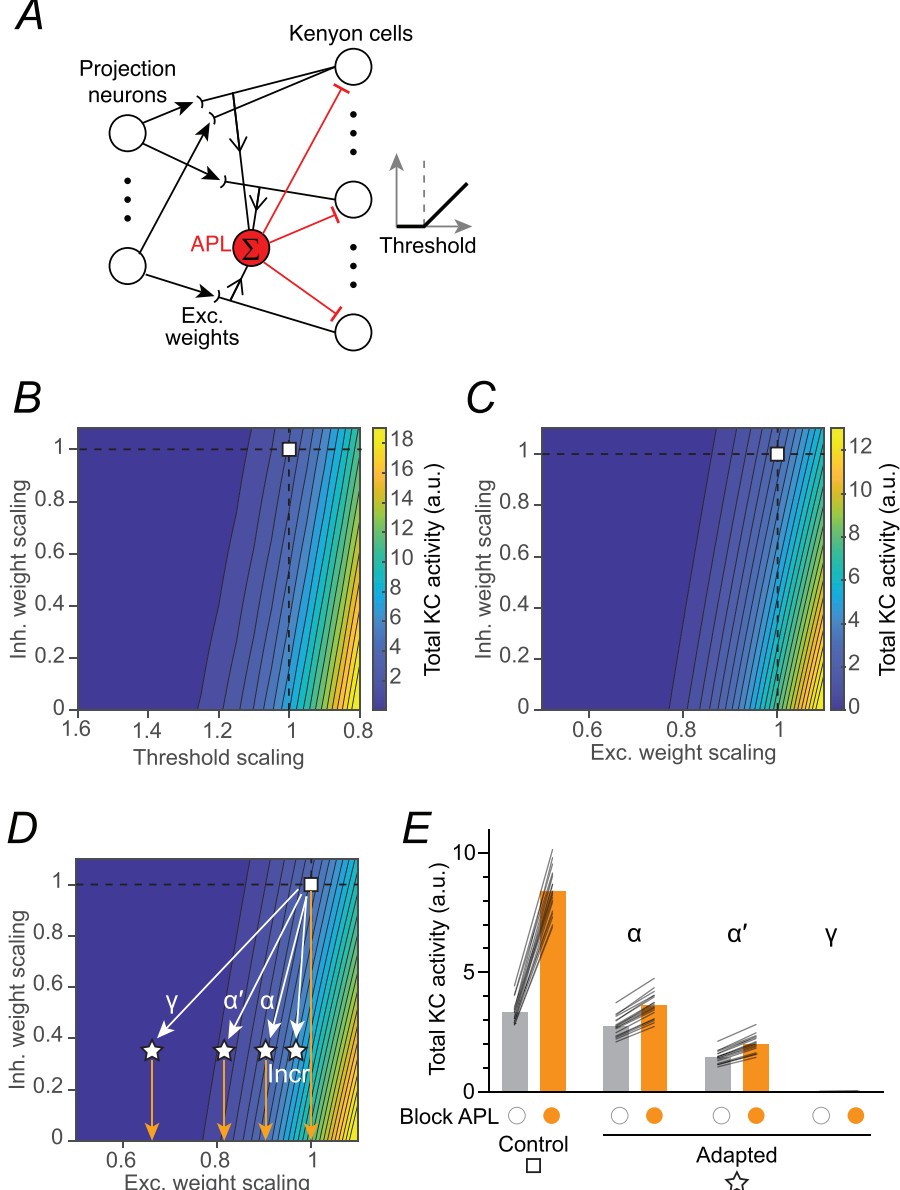

**Figure 6. Model network reproduces heterogeneous net effects of conflicting adaptations**
*A*, schematic of model. *B, C*, heat maps of total Kenyon cell (KC) activity to isoamyl acetate. Inhibitory gain, excitatory gain (*B*) and thresholds (*C*) were scaled up/down relative to the base model from Abdelrahman et al. (2021) (white square, dashed lines). Responses are averaged across $n = 20$ model instances with $n = 30$ trials of isoamyl acetate each. *D*, reducing excitation and inhibition by different amounts (moving 'southwest' on the heat map by different angles) to model possible adapted states after overactivation of KCs (stars). Orange arrows indicate the effect of blocking APL with histamine. Incr: this plasticity would increase KC activity as in Fig. 3*D* in the *α* and *α'* lobes. *E*, the parameter shifts from (*D*) qualitatively reproduce the heterogeneous effects seen experimentally in Fig. 5*C*.

Still, loss of Para is unlikely to be the entire explanation. The calyx also shows decreased excitation after adaptation (i.e. calyx responses are smaller after adaptation when APL is silenced; Fig. 5*C*) even though Para is hardly expressed in the calyx (Fig. 4) (Ravenscroft et al., 2020; Trunova et al., 2011). Although some dendritic calcium influx likely reflects backpropagating spikes (Li et al., 2013) (which would be reduced by loss of Para), most KCs show only non-spiking, subthreshold responses for any given odour (Turner et al., 2008), so much of the odour-evoked calcium influx in the calyx reflects sub-threshold responses. Future studies will reveal to what extent reduced dendritic calcium influx arises from changes to ion channels (e.g. reduced voltage-gated calcium channels) *vs.* reduced synaptic input (through pre- or postsynaptic mechanisms). These changes may be relatively modest given that our model predicts that a small reduction in synaptic input can suffice to strongly reduce KC activity (Fig. 6).

### Pathological adaptation?

In response to excess excitation, the mushroom body seems to simultaneously execute two conflicting homeo-static programmes: reducing excitation and reducing inhibition. Could this reflect second-order adaptation, where only one change occurs during KC overactivation, and the opposing change occurs during the ∼10–15 min for dissection between removing the activation and recording KC odour responses, to oppose the over-shooting rebound effect? Arguing against this inter-pretation, reduced APL sensitivity is unlikely to be second-order adaptation: APL had baseline levels of CRTC::GFP nuclear localisation after 24 h of KC over-activation (unlike the elevated nuclear localisation after 15 min or 6 h), suggesting that APL had decreased sensitivity already at the end of the 24 h when KCs were still fully activated (Fig. 1*E* and *F*). Yet the reduction in KC excitation is also unlikely to be second-order adaptation to a silent APL, as KCs do not adapt to lack of inhibition from APL (Apostolopoulou & Lin, 2020).

Rather, the most plausible explanation for the conflicting adaptations observed here is simply that the mushroom body never evolved to compensate for the particular perturbation that we imposed. Much as artificial 'super-stimulus' rewards trigger pathological reward-seeking (Jovanoski et al., 2023), it may be that neuronal perturbations outside the range encountered in nature also trigger pathological adaptations. Similarly, populations that evolved at low altitudes often show physiological adaptations to hypoxia that are beneficial at low altitudes but harmful at high altitudes (Storz & Scott, 2019). Future studies should compare adaptation to unnatural perturbations *vs.* more naturalistic perturbations, including those arising from natural variation (Abdelrahman et al., 2021). Understanding compensation for natural *vs.* unnatural perturbations will be important for defining the boundary conditions of natural homeostatic plasticity, for advancing our under-standing and treatment of pathological brain states such as epilepsy, and for engineering artificial neural circuits and brain–computer interfaces.

# Appendix

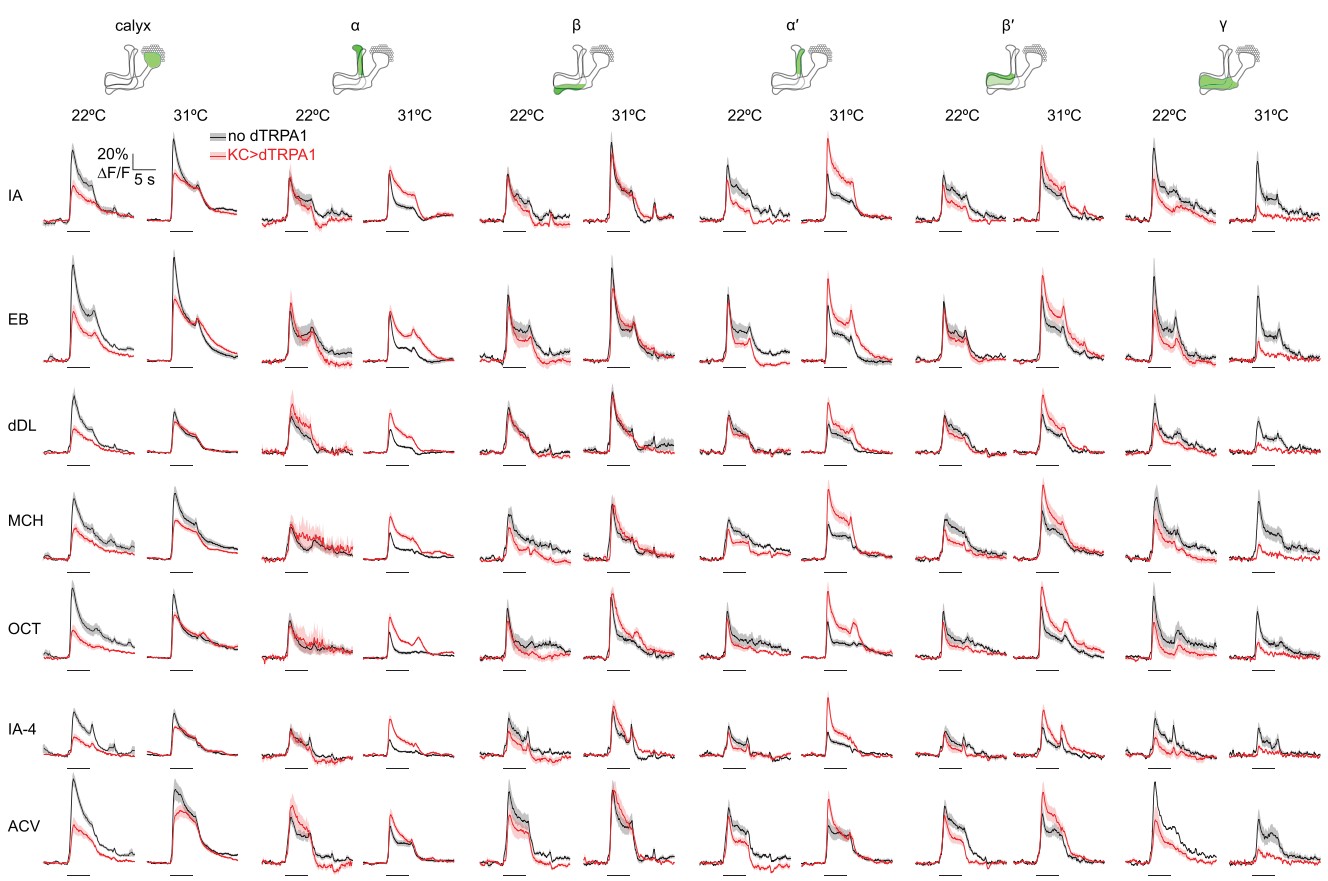

**Figure A1. Kenyon cell (KC) responses to all seven odours (related to Fig. 2 and 3)**
Traces of KC > jRGECO1a responses to seven odours [$10^{-2}$ or $10^{-4}$ isoamyl acetate (IA or IA-4), $10^{-2}$ for δ-decalactone (dDL); $10^{-1}$ ethyl butyrate (EB), $10^{-1}$ 4-methylcyclohexanol (MCH), $10^{1}$ 3-octanol (OCT) or $5*10^{-2}$ apple cider vinegar (ACV)] in KCs expressing dTRPA1 (red) or not (black), in flies kept at 22°C (left) or 31°C (right) for 24 h before the experiment, in different regions of the mushroom body as shown by the diagrams (top row). Responses to $10^{-2}$ isoamyl acetate are repeated from Fig. 3. Error shading represents SEM; horizontal line indicates the odour presentation.

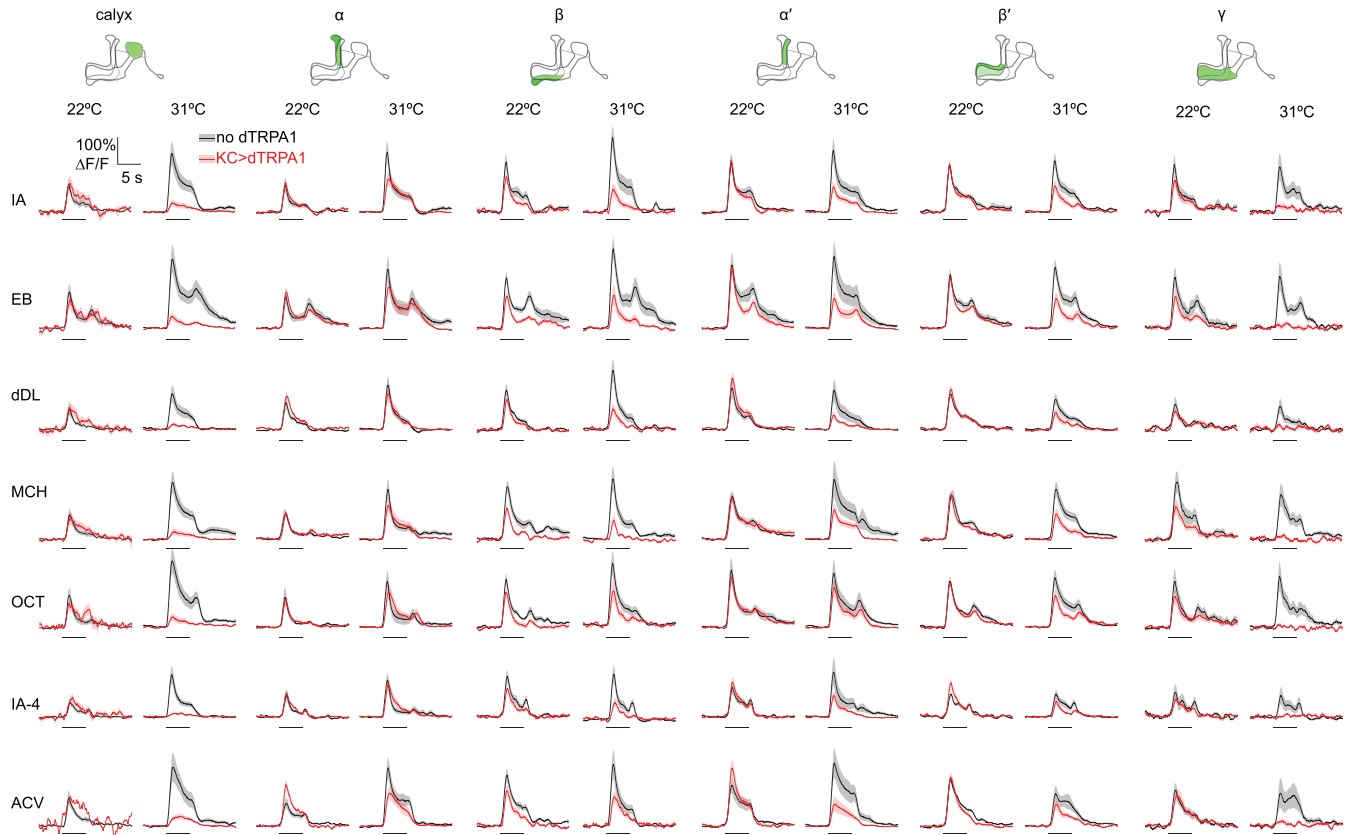

**Figure A2. Anterior paired lateral neuron (APL) responses to all seven odours (related to Fig. 2 and 3)**
Traces of APL > GCaMP6f responses to seven odours [$10^{-2}$ or $10^{-4}$ isoamyl acetate (IA or IA-4), $10^{-2}$ for
$\delta$-decalactone (dDL); $10^{-1}$ ethyl butyrate (EB), $10^{-1}$ 4-methylcyclohexanol (MCH), $10^{1}$ 3-octanol (OCT) or $5*10^{-2}$
apple cider vinegar (ACV)] in flies expressing dTRPA1 in Kenyon cells (red) or not (black), kept at 22°C (left) or
31°C (right) for 24 h before the experiment, in different regions of the mushroom body as shown by the diagrams
(top row). Responses to $10^{-2}$ isoamyl acetate in the calyx are repeated from Fig. 3. Error shading represents SEM;
horizontal line indicates the odour presentation.

*J Physiol* 604.7

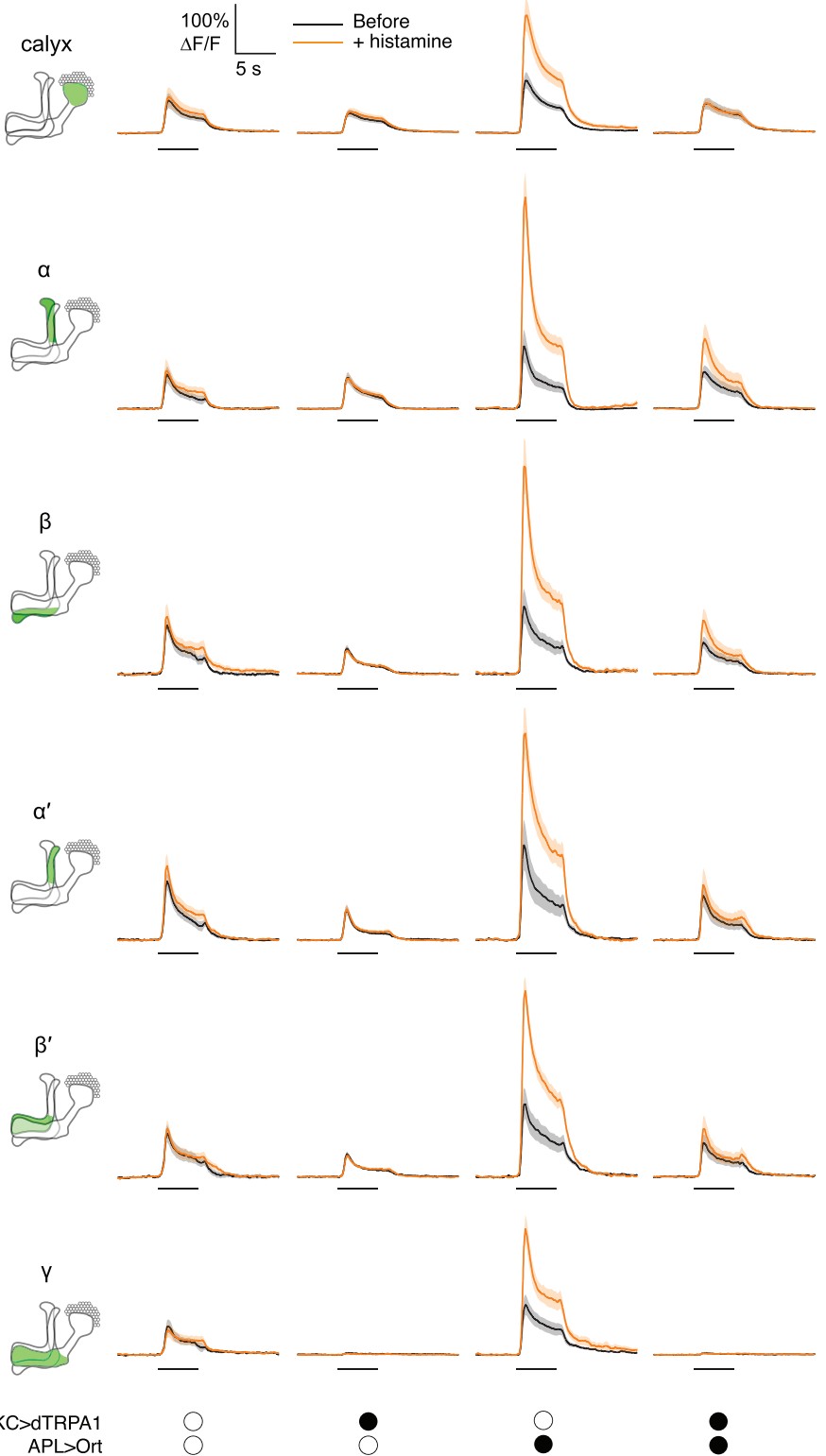

**Figure A3. All Kenyon cell (KC) responses in anterior paired lateral neuron (APL) > Ort flies (related to Fig. 5)**

Traces of KC > GCaMP6f responses to 5 s $10^{-2}$ isoamyl acetate before (black) or after (orange) histamine was applied, in brain hemispheres with neither dTRPA1 in Kenyon cells nor Ort in APL (leftmost column), KC > dTRPA1 only (second column), APL > Ort only (third column), or both KC > dTRPA1 and APL > Ort (rightmost column). The hemispheres without Ort in APL were of the same genotype as the hemispheres with Ort (indeed sometimes they were in the same fly): they merely failed to express Ort because the GH146-FLP expression is weak enough that the excision of GAL80 in tubP-FRT-GAL80-FRT only occurs ~60% of the time. These flies thus serve as genetic controls. All flies were preheated at 31°C for 24 h before imaging.

**Table A1. List of fly genotypes**

| Figure | Shorthand name/purpose | Full genotype |
|---|---|---|
| 1, 2 | KC>dTRPA1, jRGECO1a, APL>GCaMP6f | Experimental: lexAop-jRGECO1a/UAS-GCaMP6f; mb247-LexA, lexAop-dTRPA1/VT43924-GAL4.2 |
| | | Control: lexAop-jRGECO1a/UAS-GCaMP6f; mb247-LexA/VT43924-GAL4.2 |
| 1 | KC>CRTC | Experimental: UAS-CRTC::GFP, UAS-mCD8::mCherry/+; mb247-LexA, lexAop-dTRPA1/+; OK107-GAL4/+ |
| | | Control: UAS-CRTC::GFP, UAS-mCD8::mCherry/+; lexAop-dTRPA1/+; OK107-GAL4/+ |
| 1 | APL>CRTC | Experimental: UAS-CRTC::GFP, UAS-mCD8::mCherry/+; mb247-LexA, lexAop-dTRPA1/VT43924-GAL4.2 |
| | | Control: UAS-CRTC::GFP, UAS-mCD8::mCherry/+; lexAop-dTRPA1/VT43924-GAL4.2 |
| 4 | Para-FlpTag, KC>dTRPA1 | Experimental: para[FlpTag]/+; mb247-dsRed, lexAop-FLP/+; mb247-LexA, lexAop-dTRPA1/+ |
| | | Control: para[FlpTag]/+; mb247-dsRed, lexAop-FLP/+; mb247-LexA/+ |
| 5 | APL>Ort, KC>GCaMP6f, dTRPA1 | Experimental: NP2631-GAL4, GH146-FLP, UAS-mCherry/tubP-FRT-GAL80-FRT, UAS-Ort; mb247-LexA, lexAop-GCaMP6f/lexAop-dTRPA1 |
| | | Control: NP2631-GAL4, GH146-FLP, UAS-mCherry/tubP-FRT-GAL80-FRT, UAS-Ort; mb247-LexA, lexAop-GCaMP6f/+ |

**Table A2. Sources of transgenes**

| Transgene | Reference | Source |
|---|---|---|
| mb247-LexA | (Lin et al., 2014; Pitman et al., 2011) | |
| lexAop-jRGECO1a | (Dana et al., 2016) | BDSC 64426 |
| lexAop-dTRPA1 | (Burke et al., 2012; Lin et al., 2014) | |
| OK107-GAL4 | (Connolly et al., 1996) | |
| UAS-GCaMP6f (attP40) | (Chen et al., 2013) | BDSC 42747 |
| VT43924-GAL4.2 (attP2) | (Amin et al., 2020) | |
| UAS-CRTC::GFP, UAS-mCD8::mCherry/CyO; TM2/TM6B, Tb | (Bonheur et al., 2023) | BDSC 99657 |
| NP2631-GAL4 | (Tanaka et al., 2008) | Kyoto Drosophila Stock Centre, 104266 |
| GH146-FLP | (Hong et al., 2009) | |
| UAS-mCherry | (Kakihara et al., 2008) | |
| tubP-FRT-GAL80-FRT | (Gao et al., 2008) | |
| UAS-Ort | (Liu & Wilson, 2013) | Gift from Chi-hon Lee |
| lexAop-GCaMP6f | (Barnstedt et al., 2016) | |
| para[FlpTag] | (Fendl et al., 2020) | Gift from Axel Borst |
| mb247-dsRed | (Riemensperger et al., 2005) | |
| lexAop-FLP | (Pitman et al., 2011) | |

**Table A3. Details of statistical tests**

| Figure | Data | Statistical test | Comparison | *P*-value | Significance |
|---|---|---|---|---|---|
| 1*B* | KC>TRPA traces | One-sample *t* test (*vs.* zero) with Holm–Bonferroni correction | KCs rise 1 | 0.00268 | ** |
| | | | KCs fall | 0.001911 | ** |
| | | | KCs rise 2 | 0.00115 | ** |
| | | | APL rise 1 | 0.001408 | ** |
| | | | APL fall | 0.000624 | *** |
| | | | APL rise 2 | 0.003076 | ** |
| | | Two-way repeated-measures ANOVA with Geisser–Greenhouse correction | Main effect of cell type (KCs *vs.* APL) | 0.003005 | ** |
| | | | Main effect of time (rise 1, fall, rise 2) | 0.7642 | ns |
| | | | Interaction, cell type × time | 0.1232 | ns |
| | | Šídák's multiple comparisons test following two-way ANOVA | KCs rise 1 *vs.* fall | 0.5459 | ns |
| | | | KCs rise 1 *vs.* rise 2 | 0.3990 | ns |
| | | | KCs fall *vs.* rise 2 | 0.9712 | ns |
| | | | APL rise 1 *vs.* fall | 0.6061 | ns |
| | | | APL rise 1 *vs.* rise 2 | 0.9474 | ns |
| | | | APL fall *vs.* rise 2 | 0.8787 | ns |
| 1*F* | KC>CRTC | Kruskal–Wallis test | | <0.0001 | *** |
| | | Dunn's multiple comparisons test (*post hoc* test following Kruskal–Wallis test) | Control 22C *vs.* TRPA 22C | >0.9999 | ns |
| | | | TRPA 22C *vs.* TRPA 31C 15m | 0.0002 | *** |
| | | | TRPA 22C *vs.* TRPA 31C 24h | <0.0001 | *** |
| | | | Control 31C 24h *vs.* TRPA 31C 24h | <0.0001 | *** |
| | | | TRPA 31C 15m *vs.* TRPA 31C 24h | 0.2734 | ns |
| | | | Control 22C *vs.* Control 31C 24h | >0.9999 | ns |
| 1*G* | KC>CRTC cell counts | Mann–Whitney test | Control *vs.* TRPA | 0.6667 | ns |
| 1*J* | APL>CRTC | Kruskal–Wallis test | | <0.0001 | *** |
| | | Dunn's multiple comparisons test (*post hoc* test following Kruskal–Wallis test) | TRPA 22C *vs.* Control 22C | >0.9999 | ns |
| | | | TRPA 22C *vs.* TRPA 31C 15m | 0.0001 | *** |
| | | | TRPA 22C *vs.* TRPA 31C 24h | >0.9999 | ns |
| | | | Control 22C *vs.* Control 31C 24h | >0.9999 | ns |
| | | | TRPA 31C 15m *vs.* TRPA 31C 24h | <0.0001 | **** |
| | | | TRPA 31C 24h *vs.* Control 31C 24h | >0.9999 | ns |
| | | | TRPA 31C 15m *vs.* CRTL 31C 15m | 0.014 | * |
| | | | TRPA 31C 6h *vs.* Control 31C 6h | 0.0011 | ** |
| | | | TRPA 22C *vs.* TRPA 31C 6h | 0.0007 | *** |
| | | | TRPA 31C 6h *vs.* TRPA 31C 24h | <0.0001 | **** |
| | | | TRPA 31C 15m *vs.* TRPA 31C 6h | >0.9999 | ns |
| 1*L* | KCs | Mann–Whitney test | 22C *vs.* 31C | 0.9015 | ns |
| | APL | Mann–Whitney test | 22C *vs.* 31C | 0.0023 | ** |
| 2*B* | | Monte Carlo Kolmogorov–Smirnov test | Simulation *vs.* data (see Methods) | 0.71 | ns |
| 2*C* | Calyx | Kruskal–Wallis test | | 0.005 | ** |
| | | Dunn's multiple comparisons test (*post hoc* test following Kruskal–Wallis test) | Control 22C *vs.* Control 31C | 0.81 | ns |
| | | | Control 22C *vs.* TRPA 22C | 0.904 | ns |
| | | | Control 31C *vs.* TRPA 31C | 0.007 | ** |
| | | | TRPA 22C *vs.* TRPA 31C | 0.011 | * |
| | | Two-way ANOVA | Interaction temperature *vs.* genotype | 0.0004 | *** |
| 2*D* | Alpha | Kruskal–Wallis test | | 0.062 | ns |
| | | Two-way ANOVA | Interaction temperature *vs.* genotype | 0.0214 | * |

(*Continued*)

**Table A3. (Continued)**

| Figure | Data | Statistical test | Comparison | *P*-value | Significance |
|---|---|---|---|---|---|
| 2*E* | Beta | Kruskal–Wallis test | | 0.025 | * |
| | | Dunn's multiple comparisons test | Control 22C *vs.* control 31C | >0.999 | ns |
| | | (*post hoc* test following | Control 22C *vs.* TRPA 22C | >0.999 | ns |
| | | Kruskal–Wallis test) | Control 31C *vs.* TRPA 31C | 0.055 | ns |
| | | | TRPA 22C *vs.* TRPA 31C | 0.187 | ns |
| | | Two-way ANOVA | Interaction temperature *vs.* genotype | 0.2785 | ns |
| 2*F* | Alpha prime | Kruskal–Wallis test | | 0.009 | ** |
| | | Dunn's multiple comparisons test | Control 22C *vs.* control 31C | >0.999 | ns |
| | | (*post hoc* test following | Control 22C *vs.* TRPA 22C | >0.999 | ns |
| | | Kruskal–Wallis test) | Control 31C *vs.* TRPA 31C | 0.029 | * |
| | | | TRPA 22C *vs.* TRPA 31C | 0.007 | ** |
| | | Two-way ANOVA | Interaction temperature *vs.* genotype | 0.0163 | * |
| 2*G* | Beta prime | Kruskal–Wallis test | | <0.001 | *** |
| | | Dunn's multiple comparisons test | Control 22C *vs.* control 31C | >0.999 | ns |
| | | (*post hoc* test following | Control 22C *vs.* TRPA 22C | >0.999 | ns |
| | | Kruskal–Wallis test) | Control 31C *vs.* TRPA 31C | 0.021 | * |
| | | | TRPA 22C *vs.* TRPA 31C | <0.001 | *** |
| | | Two-Way ANOVA | Interaction temperature *vs.* genotype | 0.0026 | ** |
| 2*H* | Gamma | Kruskal–Wallis test | | 0.006 | ** |
| | | Dunn's multiple comparisons test | Control 22C *vs.* Control 31C | >0.999 | ns |
| | | (*post hoc* test following | Control 22C *vs.* TRPA 22C | >0.999 | ns |
| | | Kruskal–Wallis test) | Control 31C *vs.* TRPA 31C | 0.014 | * |
| | | | TRPA 22C *vs.* TRPA 31C | 0.01 | * |
| | | Two-Way ANOVA | Interaction temperature *vs.* genotype | 0.1016 | ns |
| 3*C*: APL responses | Calyx | Kruskal–Wallis test | | 0.0102 | * |
| | | Dunn's multiple comparisons test | Control 22C *vs.* TRPA 22C | >0.9999 | ns |
| | | (*post hoc* test following | Control 22C *vs.* control 31C | 0.4032 | ns |
| | | Kruskal–Wallis test) | TRPA 22C *vs.* TRPA 31C | 0.8981 | ns |
| | | | Control 31C *vs.* TRPA 31C | 0.0043 | ** |
| | | Two-way ANOVA | Interaction temperature *vs.* genotype | 0.0064 | ** |
| 3*D*: KC responses | Calyx | Ordinary one-way ANOVA | | 0.0033 | ** |
| | | Sidak's multiple comparisons test | Control 22C *vs.* TRPA 22C | 0.0604 | ns |
| | | (*post hoc* test following | Control 22C *vs.* control 31C | 0.8417 | ns |
| | | ANOVA) | TRPA 22C *vs.* TRPA 31C | 0.0735 | ns |
| | | | Control 31C *vs.* TRPA 31C | 0.3752 | ns |
| | | Two-way ANOVA | Interaction temperature *vs.* genotype | 0.2203 | ns |
| | Alpha | Welch's ANOVA | | 0.0163 | ** |
| | | Dunnett's T3 multiple comparisons test (*post hoc* test following ANOVA) | Control 22C *vs.* TRPA 22C | 0.9734 | ns |
| | | | Control 22C *vs.* control 31C | 0.9722 | ns |
| | | | TRPA 22C *vs.* TRPA 31C | 0.1493 | ns |
| | | | Control 31C *vs.* TRPA 31C | 0.0132 | * |
| | | Two-way ANOVA | Interaction temperature *vs.* genotype | 0.057 | ns |
| | Beta | Welch's ANOVA | | 0.1722 | ns |
| | | Two-way ANOVA | Interaction temperature *vs.* genotype | 0.7845 | ns |

(*Continued*)

**Table A3. (Continued)**

| Figure | Data | Statistical test | Comparison | P-value | Significance |
|--------|------|------------------|------------|---------|--------------|
| | Alpha prime | Ordinary one-way ANOVA | | <0.0001 | *** |
| | | Sidak's multiple comparisons test | Control 22C *vs.* TRPA 22C | 0.25 | ns |
| | | (*post hoc* test following | Control 22C *vs.* control 31C | 0.874 | ns |
| | | ANOVA) | TRPA 22C *vs.* TRPA 31C | <0.0001 | *** |
| | | | Control 31C *vs.* TRPA 31C | 0.0015 | ** |
| | | Two-way ANOVA | Interaction temperature *vs.* genotype | 0.000351 | *** |
| | Beta prime | Ordinary one-way ANOVA | | 0.0567 | ns |
| | | Two-way ANOVA | Interaction temperature *vs.* genotype | 0.1044 | ns |
| | Gamma | Kruskal–Wallis test | | 0.0029 | ** |
| | | Dunn's multiple comparisons test | Control 22C *vs.* TRPA 22C | 0.1766 | ns |
| | | (*post hoc* test following | Control 22C *vs.* control 31C | 0.9577 | ns |
| | | Kruskal–Wallis test) | TRPA 22C *vs.* TRPA 31C | 0.1955 | ns |
| | | | Control 31C *vs.* TRPA 31C | 0.0141 | * |
| | | Two-way ANOVA | Interaction temperature *vs.* genotype | 0.8082 | ns |
| 4D | Para | Two-way repeated-measures | Interaction segment *vs.* genotype | <0.0001 | *** |
| | | ANOVA with | Main effect of segment | 0.0008 | *** |
| | | Geisser–Greenhouse correction | Main effect of genotype | 0.0278 | * |
| | | Šídák's multiple comparisons test | Segment 1 | 0.7713 | ns |
| | | | Segment 2 | >0.9999 | ns |
| | | | Segment 3 | >0.9999 | ns |
| | | | Segment 4 | >0.9999 | ns |
| | | | Segment 5 | 0.7666 | ns |
| | | | Segment 6 | <0.0001 | **** |
| | | | Segment 7 | <0.0001 | **** |
| | | | Segment 8 | <0.0001 | **** |
| | | | Segment 9 | <0.0001 | **** |
| | | | Segment 10 | 0.0138 | * |
| | | | Segment 11 | >0.9999 | ns |
| | | | Segment 12 | 0.4946 | ns |
| | | | Segment 13 | 0.1731 | ns |
| | | | Segment 14 | 0.0927 | ns |
| | | | Segment 15 | 0.0334 | * |
| | | | Segment 16 | 0.0568 | ns |
| | | | Segment 17 | 0.7062 | ns |
| 4E | Para | Unpaired *t* test | | 0.0002 | *** |
| 5C | Calyx | Two-way repeated-measures ANOVA | Interaction histamine *vs.* genotype | 0.0005 | *** |
| | | Paired *t* test with Holm–Bonferroni correction | Control before *vs.* +histamine | 0.0027 | ** |
| | | Paired *t* test with Holm–Bonferroni correction | TRPA before *vs.* +histamine | 0.9874 | ns |
| | | Unpaired *t* test with Holm–Bonferroni correction | Before, Control *vs.* TRPA | 0.1778 | ns |
| | | Unpaired *t* test with Holm–Bonferroni correction | +Histamine, control *vs.* TRPA | 0.0008 | *** |

(*Continued*)

**Table A3. (Continued)**

| Figure | Data | Statistical test | Comparison | *P*-value | Significance |
|---|---|---|---|---|---|
| | Alpha | Two-way repeated-measures ANOVA | Interaction histamine *vs.* genotype | 0.0009 | *** |
| | | Paired *t* test with Holm–Bonferroni correction | Control before *vs.* +histamine | 0.0008 | *** |
| | | Paired *t* test with Holm–Bonferroni correction | TRPA before *vs.* +histamine | 0.0332 | * |
| | | Mann–Whitney test with Holm–Bonferroni correction | Before, control *vs.* TRPA | 0.5573 | ns |
| | | Unpaired *t* test with Holm–Bonferroni correction | +Histamine, control *vs.* TRPA | 0.006 | ** |
| | Beta | Two-way repeated-measures ANOVA | Interaction histamine *vs.* genotype | 0.001 | ** |
| | | Paired *t* test with Holm–Bonferroni correction | Control before *vs.* +histamine | 0.0033 | ** |
| | | Wilcoxon test with Holm–Bonferroni correction | TRPA before *vs.* +histamine | 0.0196 | * |
| | | Mann–Whitney test with Holm–Bonferroni correction | Before, control *vs.* TRPA | 0.063 | ns |
| | | Unpaired *t* test with Holm–Bonferroni correction | +Histamine, control *vs.* TRPA | 0.0004 | *** |
| | Alpha prime | Two-way repeated-measures ANOVA | Interaction histamine *vs.* genotype | 0.0001 | *** |
| | | Paired *t* test with Holm–Bonferroni correction | Control before *vs.* +histamine | 0.0006 | *** |
| | | Wilcoxon test with Holm–Bonferroni correction | TRPA before *vs.* +histamine | 0.2754 | ns |
| | | Mann–Whitney test with Holm–Bonferroni correction | Before, control *vs.* TRPA | 0.229 | ns |
| | | Unpaired *t* test with Holm–Bonferroni correction | +Histamine, Control *vs.* TRPA | 0.0004 | *** |
| | Beta prime | Two-way repeated-measures ANOVA | Interaction histamine *vs.* genotype | <0.0001 | *** |
| | | Paired *t* test with Holm–Bonferroni correction | Control before *vs.* +histamine | <0.0001 | *** |
| | | Paired *t* test with Holm–Bonferroni correction | TRPA before *vs.* +histamine | 0.0915 | ns |
| | | Mann–Whitney test with Holm–Bonferroni correction | Before, control *vs.* TRPA | 0.0576 | ns |
| | | Unpaired *t* test with Holm–Bonferroni correction | +Histamine, control *vs.* TRPA | <0.0001 | *** |
| | Gamma | Two-way repeated-measures ANOVA | Interaction histamine *vs.* genotype | 0.0159 | * |
| | | Paired *t* test with Holm–Bonferroni correction | Control before *vs.* +histamine | 0.052 | ns |
| | | Paired *t* test with Holm–Bonferroni correction | TRPA before *vs.* +histamine | 0.964 | ns |
| | | Mann–Whitney test with Holm–Bonferroni correction | Before, control *vs.* TRPA | <0.0001 | *** |
| | | Unpaired *t* test with Holm–Bonferroni correction | +Histamine, control *vs.* TRPA | <0.0001 | *** |

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

## Additional information

### Data availability statement

All data necessary to reproduce our findings and figures are included in Dataset S1. Analysis code is available on GitHub at https://github.com/aclinlab/calcium-imaging and https://github.com/aclinlab/bergmann-et-al.

### Competing interests

The authors declare no competing interests.

### Author contributions

G.A.B.: conceptualisation, methodology, software, investigation, visualisation, writing – original draft, writing – review and editing; M.W.T.: methodology, investigation, writing – review and editing; K.G.-W.: methodology, investigation, writing – review and editing; P.J.F.: investigation, formal analysis, writing – original draft, writing – review and editing, visualisation; T.C.C.: investigation, writing – review and editing; A.C.L.: conceptualisation, methodology, software, formal analysis, writing – original draft, writing – review and editing, visualisation, supervision, funding acquisition. All authors approved the final version of the manuscript and agreed to be accountable for all aspects of the work in ensuring that questions related to the accuracy or integrity of any part of the work are appropriately investigated and resolved. All persons designated as authors qualify for authorship, and all those who qualify for authorship are listed.

## Funding

This work was supported by the European Research Council (639489), the Wellcome Trust (225814/Z/22/Z), the Biotechnology and Biological Sciences Research Council (BB/X000273/1, BB/X014568/1, BB/S016031/1) and the BBSRC White Rose Doctoral Training Partnership (BB/M011151/1, BB/T007222/1).

## Acknowledgements

We thank members of the Lin lab for discussion; Moshe Parnas and Anton Nikolaev for comments on the manuscript; the Wolfson Light Microscopy Facility for support with confocal imaging; Kath Whitley, Kate Fewkes, Cherry O'Keefe, Eleanor James, Rachid Achour and Aaron Kirkby-Nowogorski for technical assistance; and Axel Borst (MPI Munich) and the Bloomington Stock Centre for flies. For the purpose of open access we have applied a Creative Commons Attribution (CC BY) licence to any Author Accepted Manuscript version arising from this submission.

## Author's present address

Gregor A. Bergmann: Institute for Anatomy and Cell Biology, Heidelberg University, Im Neuenheimer Feld 307, 69120 Heidelberg, Germany.

## Keywords

*Drosophila*, feedback inhibition, homeostatic plasticity, mushroom body, olfaction

## Supporting information

Additional supporting information can be found online in the Supporting Information section at the end of the HTML view of the article. Supporting information files available:

**Peer review history**
**Dataset S1**

