## [Peer review history · The Journal of Physiology]

Conflicting adaptations in an inhibitory feedback circuit

Gregor A Bergmann, Melissa W Tan, Katie Greenin-Whitehead, Philippe J Fischer, Thomas C Cozens, and Andrew C Lin
DOI: 10.1113/JP290394

Corresponding author(s): Andrew Lin (andrew.lin@sheffield.ac.uk)

Review Timeline:	Submission Date:	27-Oct-2025
	Editorial Decision:	08-Dec-2025
	Revision Received:	22-Jan-2026
	Accepted:	04-Feb-2026

Senior Editor: Nathan Schoppa

Reviewing Editor: Nathan Schoppa

Transaction Report:

Re: JP-RP-2025-290394 "Conflicting adaptations in an inhibitory feedback circuit" by Gregor A Bergmann, Melissa W Tan, Katie Greenin-Whitehead, Philippe J Fischer, Thomas C Cozens, and Andrew C Lin

Dear Dr Lin,

Thank you for submitting your manuscript to The Journal of Physiology. It has been assessed by a Reviewing Editor and by 2 expert referees and we are pleased to tell you that it is potentially acceptable for publication following satisfactory major revision.

Please address all the points raised and incorporate all requested revisions or explain in your Response to Referees why a change has not been made. We hope you will find the comments helpful and that you will be able to return your revised manuscript within 2 months. If your article is NOT for a Special Issue, you may have 9 months to revise. If you require an extension, please contact journal staff: jp@physoc.org. Please note that this letter does not constitute a guarantee for acceptance of your revised manuscript.

REVISION CHECKLIST:

We look forward to receiving your revised submission.

Yours sincerely,

Nathan Schoppa
Senior Editor
The Journal of Physiology

REQUIRED ITEMS

- The contact information for the person responsible for 'Research Governance' at your institution needs to be provided. This includes their name and an institutional email address. Please ensure the contact is not an author on this paper and provide an alternate contact if necessary, or confirm in the submission form that the author whose email was provided has sole responsibility for research governance. This is the person who is responsible for regulations, principles and standards of good practice in research carried out at the institution, for instance the ethical treatment of animals, the keeping of proper experimental records or the reporting of results.

- Please upload separate high-quality figure files via the submission form.

- Please ensure that the Article File you upload is a Word file.

EDITOR COMMENTS

Senior Editor:

Comments to the Author:

Your manuscript has been reviewed by two expert reviewers, who both felt that the work is addressing an interesting, unresolved question about mechanisms of activity-dependent homeostasis, i.e. whether systems attempt to stabilize the set-point of individual neurons versus the network. This study is examining this question in the *Drosophila* olfactory system, but it was felt that the results would have implications across different models. Both reviewers were also quite enthusiastic about many aspects of the experimental results and analyses performed, finding most of the work convincing and thorough. A number of significant issues were however raised that would need to be addressed in a revised manuscript. Amongst these are:

1. Reviewer 1 wonders about the precise locus of plasticity in the KC-to-APL connections. Experiments to address this point are likely beyond the scope of this study (as the reviewer acknowledges), but the authors should consider interpreting their results more conservatively in the absence of this information.

2. Reviewer 2 raises a number questions about the statistical analyses performed.

3. The authors should improve the images shown in Fig. 1 to make statements about specific expression of genes more

convincing and also provide more information about controls used for their probes. Also related to Fig. 1, clarification is needed on how KC and APL response times were quantified.

4. Data traces that support Fig 2 that are now in a supplementary figure should be moved to the main figure and clarification should be provided about how these results were quantified.

5. Some clarification in the presentation of other figures and methods are required. This is needed in part to account for the fact that most readers of the article will not be experts in *Drosophila* genetics.

6. The manuscript needs to be reformatted for *Journal of Physiology*, for example by moving the Methods toward the front end. Also, the authors should understand that *Journal of Physiology* does not publish supplementary figures or tables in the final publication, though such figures/tables are allowed in the review stage. The authors should keep this in mind as they consider what information might be moved from supplementary to main figures/tables. Clearly, the supplementary table showing p-values needs to be included as a regular table.

7. Minor point: the legend of Fig. 2 should make it clear that exact p-values are provided in a table.

REFEREE COMMENTS

Referee #1:

This manuscript reports the results of an expertly performed and analysed set of experiments on activity-dependent plasticity in the *Drosophila* olfactory system. It tackles an interesting and important question in neuronal plasticity, on how multiple forms of compensatory plasticity can interact to influence neuronal response properties. In this case, the authors present an intriguing scenario where activation of one cell type for 24h produces different effects on (likely) intrinsic excitability and feedback inhibition, producing no consistent alteration in sensory responses. The data from all experiments appear thoroughly convincing, are accurately and comprehensively described, and are interpreted with some appropriate and interesting discussion. I enjoyed reading this, and only have a few suggestions for potential improvement:

Major issue

The plastic changes underlying all of the observed alterations in response properties after 24h KC activation could take place exclusively within the KCs themselves. This would be the case if the reduced activation of APL is entirely due to altered presynaptic properties at KC->APL connections (mentioned as a possibility in the Discussion). If this is true, then there's no issue here of cell- vs network-level adaptation in inhibitory neurons, a theoretical conflict which is currently used to frame this study (Fig 1A). Instead, there would be a situation where excitatory neurons dial down multiple aspects of their function in response to over-activation, with no ability to reduce output selectively to specific downstream partners. It would be really helpful to have data on the precise locus of plasticity to resolve this, but I appreciate that this is a substantial undertaking that forms the focus of future work. In the absence of any such localisation, the authors need to be very careful in framing and interpreting the effects here, which are still absolutely 'conflicting', but could be the result of multiple output-independent changes happening in excitatory neurons rather than any specific forms of plasticity in inhibitory neurons per se.

Minor suggestions

I know calcium imaging data have very limited temporal resolution, but perhaps the authors could try looking at response onset dynamics. A lack of feedback inhibition should only be apparent after a certain delay from response onset, so perhaps KC odour responses after 24h activation start off much more weakly than controls, before a relatively lack of feedback inhibition allows them to grow to similar plateau levels?

In my mind the really interesting variability in this study is found in Fig3D. Can the model (Fig 6) be used to explain this variability, rather than the rather consistent effects observed in Fig5C?

Could there be a decrease in the length of the para-defined segments in 24h-activated KCs, as well as a decrease in labelling intensity? If so, this would fit with activity-dependent AIS shortening observed in many mammalian models, and could additionally contribute to decreased KC intrinsic excitability.

Referee #2:

The study by Bergmann and colleagues tackle a core question in neurophysiology: when homeostatic plasticity is at play, which set point is it aiming to stabilize, the one of the individual neuron, or the one of the network? This is potentially conflicting for inhibitory cells, who may adopt opposing plasticity mechanisms depending on the answer to the above question. The experimental setup with the fruit fly is robust, and while the findings are generalizable to other model system, it is at time difficult to follow the various genetic lines and tools for non-drosophila experts. The results are novel and surprising but the thoroughness of the experimental design and the depth at which the authors pushed the analysis is such that it supports them - especially the killer experiment presented in figure 5, which is the highlight of the paper. Without a doubt, they will spark a much-needed debate about how homeostasis plays out to support cellular and network needs. Therefore, publication of such a timely and well-designed study in the Journal of Physiology is certainly warranted, albeit after addressing a few concerns listed below.

Major points:

- The manuscript is generally well written. However, in trying hard to pre-empt reader confusion, the text sometimes becomes defensive and overly guided, often telling the reader what to expect or how to interpret results before the data are presented. This can make the narrative feel convoluted rather than clear, especially for those not deeply familiar with Drosophila genetics/tools. A fresh read-through with an eye toward reducing this defensiveness would be recommended.
- Fig 4 fails to fully convince that more fluorescence really means more functional sodium channels. This is aggravated by the very convoluted quantification in fig 4D (took three re-reads and many attempt at sketching the process to perhaps understand it) and questionable statistics that accompanies the claim. Please add a better schematic of the analysis and explain it better, in the methods. Moreover, it seems like it is an artificial way to do an unnested statistics. Similarly, if the grasp of what happen is correct, the stats should be nested, multiple comparisons, and repeated measures.
- Fig 1B: Kc and APL are not responding in the same way at the beginning of their response, but this is not acknowledged nor quantified. It likely should, because this has implications for the interpretation of following results. Please quantify and show that it is different from baseline (aka define criteria to say when a cell responds) and to each other.
- While the table summarizing the statical analysis is useful, the statistical approach is not consistent in paper, especially the fact that repeated measures and paired tests are not done throughout when needed (see fig 1F, 4E)
- Lacking the necessary computational skill to properly comment on the model, the only thing to add here is that it is a shame not to use it to make predictions, for instance of what would happen for more protracted / more naturalistic types of stimuli. If feasible (and mind you, it may not be) this could tie in very well with the evolutionary aspects to natural vs unnatural perturbations presented in the discussion (lines 355 onwards).
- In figure 1, the example images shown make it impossible to appreciate that the various genetically expressed things are indeed expressed in the correct cells because everything is extremely zoomed in and not annotated. This is essential to assess the entire paper. Please modify the figure and ideally show control experiments of specificity and penetrance of expression, e.g. with staining.
- General point about figures: ideally, they should all be as clear and easy to follow as fig 5 with schematic of experimental design, example traces, and quantification of such traces. For example, Figs. 1a and 2g present the same information, so from a clarity and storytelling prospective it is unclear why they are not combined into a single figure. In addition, Fig. 1a is not a schematic of the experiment but rather a hypothesis for the entire study, which makes its placement within an already complex figure confusing. Furthermore, Fig. 2 does not include a single example trace to accompany the quantification; all traces are instead placed in the supplementary material. Given this, it is difficult for the reader to understand exactly what is being quantified in Fig. 2. At minimum, a representative example trace should be included in the main figure, with the remaining examples left in the supplementary material.

Minor comments:

- The introduction feels too long and even a bit patronizing. While it is understandable that the authors want to set the scene for what are unexpected results later on, they may want to rein it in a little. Instead, they could use the save words to help the non drosophilologist understand better the various genetic lines use as well as the temperature manipulation that is really not common in mammalian work.
- Along the same lines, please explain more clearly why the various regions they are recording from are relevant/different to contextualize both fig 2 and 3 for non-drosophila experts.
- Why using apple cider vinegar from a supermarket together with a panel of monomolecular odours from sigma? Was there a hypothesis of seeing different things? Moreover, why are the monomolecular odours naturalistic?
- Please annotate images on 4b to orient the reader. While colour choice is commendable since it caters for colour blind colleagues, green and purple merge into massive white smears that looks like noise instead of the interesting signal.
- Please add figure legend for A1 and A2 instead of just mentioning them in the legend of figure 2.
- Discussion, line 343: please check the analogy with air conditioning and heating, it does not seem to mirror what you are saying for excitation and inhibition.

END OF COMMENTS

Response to reviewers, manuscript JP-RP-2025-290394

We thank the editor and reviewers for the constructive comments which have greatly improved the manuscript. Line numbers below are given as (line number in plain manuscript / line number in tracked changes version).

EDITOR COMMENTS

Senior Editor:

Comments to the Author:

Your manuscript has been reviewed by two expert reviewers, who both felt that the work is addressing an interesting, unresolved question about mechanisms of activity-dependent homeostasis, i.e. whether systems attempt to stabilize the set-point of individual neurons versus the network. This study is examining this question in the Drosophila olfactory system, but it was felt that the results would have implications across different models. Both reviewers were also quite enthusiastic about many aspects of the experimental results and analyses performed, finding most of the work convincing and thorough. A number of significant issues were however raised that would need to be addressed in a revised manuscript. Amongst these are:

1. Reviewer 1 wonders about the precise locus of plasticity in the KC-to-APL connections. Experiments to address this point are likely beyond the scope of this study (as the reviewer acknowledges), but the authors should consider interpreting their results more conservatively in the absence of this information.

We have revised the framing of the manuscript to be more conservative (focussing more on “conflicting” adaptations, and referring to “local” vs. “network” adaptation as opposed to “neuron-centered” vs. “network-centered”).

2. Reviewer 2 raises a number questions about the statistical analyses performed.

We have clarified our statistical analyses and corrected an omission in the statistics table.

3. The authors should improve the images shown in Fig. 1 to make statements about specific expression of genes more convincing and also provide more information about controls used for their probes. Also related to Fig. 1, clarification is needed on how KC and APL response times were quantified.

We have added new zoomed out images of KCs and APL expressing CRTC::GFP and a negative control with UAS-CRTC::GFP (no GAL4) also stained with anti-GFP.

We have added quantification of the amplitude of KC and APL responses to TRPA heating in new Fig. 1B,C. We present quantification of response times in the response to Reviewer 2 below but have not included them in the manuscript as we do not believe they are particularly interesting.

4. Data traces that support Fig 2 that are now in a supplementary figure should be moved to the main figure and clarification should be provided about how these results were quantified.

We included the raw traces from which the example scatter plot was derived as a new panel 2B, with a shaded region to indicate the period over which the mean $\Delta F/F$ is calculated. We also added

“quantified the responses as the mean $\Delta F/F$ during the 5 s odour stimulus” to the main text at line 394 / 438. As explained in point 6 below, we would prefer to keep the full collection of all average data traces in Fig. A1 and A2 in the Appendix.

5. Some clarification in the presentation of other figures and methods are required. This is needed in part to account for the fact that most readers of the article will not be experts in Drosophila genetics.

We have removed the original Fig. 1A, redrawn the schematic in Fig. 4C, clarified the methods for the ParaFlpTag quantification, and clarified the Results text about Drosophila genetics.

6. The manuscript needs to be reformatted for Journal of Physiology, for example by moving the Methods toward the front end. Also, the authors should understand that Journal of Physiology does not publish supplementary figures or tables in the final publication, though such figures/tables are allowed in the review stage. The authors should keep this in mind as they consider what information might be moved from supplementary to main figures/tables. Clearly, the supplementary table showing p-values needs to be included as a regular table.

We have reformatted the manuscript. The Journal of Physiology’s Information for Authors (https://jp.msubmit.net/cgi-bin/main.plex?form_type=display_requirements#suppinfo) states about supporting information that:

If inclusion of this material would interrupt the ‘flow’ of the paper, authors are welcome to include this additional material as an Appendix at the end of the article (rather than as separate supporting information). An Appendix should be included within the article file of a revised version; any figures and tables should be numbered as Fig. A1, Fig. A2, Table A1, etc. – and each should be cited in the text.

We felt that the sheer volume of traces in our supplementary figures is overwhelming and interrupts the flow of the paper, so we would like to include them in the Appendix as Fig. A1, A2, etc. We have included the table of p-values as Table A3.

A note about the statistics policy, which requires that error bars show SD rather than SEM: our error bars show the 95% confidence interval (CI) of the mean. We feel that mean \pm 95% CI is the most appropriate summary statistic (as opposed to mean \pm SD) because:

- The 95% CI gives an interval where the true mean is likely to fall, with 95% confidence (more precisely: if we repeatedly sample from the same underlying population, and calculate the 95% CI for each sample, then the true mean would fall inside that interval in ~95% of samples). We feel this is a more meaningful metric to show than SD when comparing the means of different groups, because it means the overlap in error bars provides a reasonable (though obviously not foolproof) visual impression of whether differences between groups are likely to be statistically significant.
- Since we show individual data points, the full spread of variability in data is already visible, so showing the SD adds less information than showing the 95% CI.
- We understand that people often choose to show SEM over SD because it makes the error bars smaller; however, 95% CI is actually bigger than the SD when $n \leq 6$.

We do use SEM for the calcium traces, but their corresponding data summaries show the individual data points with 95% CI. We do not use the SEM alone for any data summaries.

7. Minor point: the legend of Fig. 2 should make it clear that exact p-values are provided in a table.

We have added the phrase “and exact p-values” in all figure legends: “see **Table A3** for detailed statistics and exact p-values”.

REFEREE COMMENTS

Referee #1:

This manuscript reports the results of an expertly performed and analysed set of experiments on activity-dependent plasticity in the Drosophila olfactory system. It tackles an interesting and important question in neuronal plasticity, on how multiple forms of compensatory plasticity can interact to influence neuronal response properties. In this case, the authors present an intriguing scenario where activation of one cell type for 24h produces different effects on (likely) intrinsic excitability and feedback inhibition, producing no consistent alteration in sensory responses. The data from all experiments appear thoroughly convincing, are accurately and comprehensively described, and are interpreted with some appropriate and interesting discussion. I enjoyed reading this, and only have a few suggestions for potential improvement:

Major issue

The plastic changes underlying all of the observed alterations in response properties after 24h KC activation could take place exclusively within the KCs themselves. This would be the case if the reduced activation of APL is entirely due to altered presynaptic properties at KC->APL connections (mentioned as a possibility in the Discussion). If this is true, then there's no issue here of cell- vs network-level adaptation in inhibitory neurons, a theoretical conflict which is currently used to frame this study (Fig 1A). Instead, there would be a situation where excitatory neurons dial down multiple aspects of their function in response to over-activation, with no ability to reduce output selectively to specific downstream partners. It would be really helpful to have data on the precise locus of plasticity to resolve this, but I appreciate that this is a substantial undertaking that forms the focus of future work. In the absence of any such localisation, the authors need to be very careful in framing and interpreting the effects here, which are still absolutely 'conflicting', but could be the result of multiple output-independent changes happening in excitatory neurons rather than any specific forms of plasticity in inhibitory neurons per se.

We agree with the reviewer and we are keen to address this important question in future work. We have re-framed the paper more carefully to focus on the idea of 'conflicting' adaptations, and we changed the phrasing from "neuron-centered" vs. "network-centered" to "local" vs. "network". We believe this theoretical framing is more in keeping with our evidence, because non-specific changes in excitatory output would still be a kind of "local" homeostasis (here the controlled variable might be, e.g., synaptic release), which contradicts homeostasis of neuronal activity at the wider network level.

The key new passage in the Introduction reads:

Depending on which variables are being homeostatically controlled, homeostatic changes in different parts of a network might oppose each other. For example, if excitatory neurons are overly active, they should decrease their excitability. In inhibitory feedback networks, this decrease can be supported by a homeostatic increase in feedback inhibition (Sachse et al., 2007; Das et al., 2011; Kuhlman et al., 2013), e.g. via increased excitability of inhibitory neurons (Chand et al., 2015; Gainey et al., 2018) or stronger excitatory-to-inhibitory synapses (Chang et al., 2010). However, the opposite might occur if more "local" variables

are homeostatically controlled, e.g., if inhibitory neurons stabilize their own activity (rather than their total excitatory input), or if excitatory-to-inhibitory presynaptic terminals stabilize their average synaptic output (rather than the activity of the excitatory neuron). In these cases, excess activity of excitatory neurons would lead to *decreased* activity of inhibitory neurons, which would be expected to make excitatory neurons *even more* active, thus opposing homeostasis at the network level. Such “local” homeostasis of inhibitory activity has been observed in dissociated neurons and brain slices (Desai et al., 1999; Gibson et al., 2006; Bartley et al., 2008; Wenner, 2011). However, it remains unclear whether such local inhibitory homeostasis occurs in the intact central brain in vivo and whether it actually causes anti-homeostatic effects on the activity of excitatory neurons.

(lines 74-91 / 79-106) [Line numbers are given as (line number in plain manuscript / line number in tracked changes version).]

We rephrased the Abstract and Key Points to mention reducing inhibitory activity as just one example of a kind of “local” adaptation that could conflict with “network” adaptation:

Abstract:

However, homeostatic mechanisms operating at different levels may conflict with each other. For example, in inhibitory feedback circuits, if inhibitory neurons receive excess excitation, compensation at a “local” level (e.g., reducing inhibitory neurons’ activity) could conflict with “network-level” compensation (e.g., suppressing the excitatory neurons responsible for over-exciting the inhibitory neurons).

Key Points (1,2,5):

- Neural networks maintain stable activity levels through homeostatic plasticity – but **whose activity levels** what physiological variables are stabilized?
- In inhibitory feedback circuits, ~~if inhibitory neurons stabilize their own activity, this might conflict with network-level compensation~~ local and network-level compensation might conflict. For example, if excitatory neurons are over-active, they might compensate by becoming less excitable. But if inhibitory neurons compensate for the excess excitation by also becoming less excitable, this would decrease inhibition onto the excitatory neurons and increase their activity.
- These results show that adaptation at **neuronal-local** and network levels can conflict with each other.

Minor suggestions

I know calcium imaging data have very limited temporal resolution, but perhaps the authors could try looking at response onset dynamics. A lack of feedback inhibition should only be apparent after a certain delay from response onset, so perhaps KC odour responses after 24h activation start off much more weakly than controls, before a relatively lack of feedback inhibition allows them to grow to similar plateau levels?

We thank the reviewer for this interesting suggestion. We investigated the response onset dynamics by quantifying the response latency as the time at which the trace crossed half its maximum value. We also qualitatively examined the shape of the initial upward slope of fluorescence. We did not find consistent differences between control and adapted Kenyon cells by either measure.

However, we also found no differences in latency or shape of upward slope when feedback inhibition was completely abolished by APL>Ort+histamine in un-adapted flies (data in Fig. 5). It can also be seen in the black vs. orange traces in Fig. 5 that adding histamine doesn't change the overall shape of the odour responses; they are simply scaled up. (The same can be seen in our previously published results when APL was blocked by tetanus toxin or Ort: (Apostolopoulou & Lin, 2020)). If loss of feedback inhibition resulted in a delayed increase in response, we would have expected to see a slower rise to peak (i.e., the trace would reach half its maximum value later), or a flatter overall response (i.e., higher plateau relative to the peak). Neither occurred. Thus, it seems that loss of feedback inhibition per se does not alter response onset dynamics, at least as measured by calcium imaging, so the fact that we didn't see altered response onset dynamics after KC activation doesn't provide evidence one way or the other about whether they have lost feedback inhibition.

It may not be surprising that losing feedback inhibition doesn't cause obvious changes in response onset dynamics. Much of APL's feedback inhibition occurs locally on KC dendrites via KC->APL synapses within the calyx. KCs release acetylcholine from their dendrites (Christiansen et al., 2011) and most likely can do so without spiking (as depolarization and Ca²⁺ influx through nAChRs is likely sufficient to trigger dendritic ACh release from KCs). APL itself is non-spiking and provides feedback to KCs locally (Amin et al., 2020). Thus, feedback inhibition from APL (or the effects of lack of inhibition) may well occur with little delay, accounting for our inability to resolve differences in response dynamics in our current calcium imaging data.

In my mind the really interesting variability in this study is found in Fig3D. Can the model (Fig 6) be used to explain this variability, rather than the rather consistent effects observed in Fig5C?

Yes, it can. Indeed, the star marked "Incr" on Fig. 5D was meant to indicate how the model can reproduce the increased Kenyon cell seen in the α and α' lobes in Fig. 3D. We apologise for not making this clear enough in the original manuscript and have emphasised this point by the following edit on the main text (lines 538-542 / 586-591):

In particular, by scaling excitation differently while scaling inhibition by a constant amount, we could reproduce the experimentally observed heterogeneous dTRPA1-induced changes in the odor responses of $\alpha\beta$, $\alpha'\beta'$ and γ Kenyon cells (**Figure 6E**, compare to **Figure 5C**). **The increased responses in the α and α' lobe in Figure 3D could also be reproduced (star marked 'Incr' in Figure 6D).**

Could there be a decrease in the length of the para-defined segments in 24h-activated KCs, as well as a decrease in labelling intensity? If so, this would fit with activity-dependent AIS shortening observed in many mammalian models, and could additionally contribute to decreased KC intrinsic excitability.

We wondered this as well and indeed this was part of our original motivation in carrying out the Para labelling experiments. However, the Para signal we observed did not have clear enough "edges" to allow us to define the length of the AIS. It can be seen in the example images in Fig. 4 that, while the Para fluorescence "begins" fairly suddenly at the boundary between the calyx and peduncle, the Para doesn't suddenly disappear in the distal axon; rather, it fades away gradually, and there is still more Para in the distal axon than in the dendrites - which makes sense given that presumably the axons still need to conduct action potentials out to the distal tip of the axon. Given this lack of clear edges, we felt it was more appropriate to simply present the raw data of the Para

signal at each segment. To avoid giving the reader the impression that we can quantify the boundaries of the AIS, we removed the AIS shading from the schematic in Figure 4C.

Referee #2:

The study by Bergmann and colleagues tackle a core question in neurophysiology: when homeostatic plasticity is at play, which set point is it aiming to stabilize, the one of the individual neuron, or the one of the network? This is potentially conflicting for inhibitory cells, who may adopt opposing plasticity mechanisms depending on the answer to the above question. The experimental setup with the fruit fly is robust, and while the findings are generalizable to other model system, it is at time difficult to follow the various genetic lines and tools for non-drosophila experts. The results are novel and surprising but the thoroughness of the experimental design and the depth at which the authors pushed the analysis is such that it supports them - especially the killer experiment presented in figure 5, which is the highlight of the paper. Without a doubt, they will spark a much-needed debate about how homeostasis plays out to support cellular and network needs. Therefore, publication of such a timely and well-designed study in the Journal of Physiology is certainly warranted, albeit after addressing a few concerns listed below.

Major points:

- The manuscript is generally well written. However, in trying hard to pre-empt reader confusion, the text sometimes becomes defensive and overly guided, often telling the reader what to expect or how to interpret results before the data are presented. This can make the narrative feel convoluted rather than clear, especially for those not deeply familiar with Drosophila genetics/tools. A fresh read-through with an eye toward reducing this defensiveness would be recommended.*

We thank the reviewer for this feedback. We have read through the manuscript with an eye toward reducing defensiveness but struggled to find specific changes to make. We suspect the “convoluted” feeling arises because the subject is inherently somewhat convoluted, as we observe two opposing adaptations leading to heterogeneous net effects. In previous drafts of the manuscript, we presented the results with less “preamble” explaining predictions of various alternative hypotheses, and explained our interpretation afterward, but we found that it was difficult for readers to understand the results without having some theoretical framework presented first, to slot the results into. (Such passages include the preambles to Figs. 3 and 5.) Weighing the relative risks of over-explaining vs. leaving the reader confused, we feel that risking over-explaining is the lesser of two evils. We have made clarifying edits about Drosophila genetics (see Minor points below).

- Fig 4 fails to fully convince that more fluorescence really means more functional sodium channels. This is aggravated by the very convoluted quantification in fig 4D (took three re-reads and many attempt at sketching the process to perhaps understand it) and questionable statistics that accompanies the claim. Please add a better schematic of the analysis and explain it better, in the methods.*

We have clarified the Methods text by rearranging and rewording the text and adding subheadings (lines 212-248 / 244-288). [Line numbers are given as (line number in plain manuscript / line number in tracked changes version).] We have also modified the schematic in Figure 4C by removing the AIS (which we felt was a distraction; see also Reviewer 1’s last point) and drawing the evenly spaced nodes and Voronoi cell divisions on the upper panel cartoon of the mushroom body anatomy (not just on the lower panel’s series of squares).

We have edited the Discussion to be more careful not to imply that decreased Para expression necessarily means decreased voltage-gated sodium conductance (line 565-568 / 613-616):

This loss of excitation may be attributable in part to the reduced expression of Para voltage-gated Na⁺ channels (Figure 4); **if this translates to as** a reduced voltage-gated Na⁺ conductance, **that** should make it less likely for Kenyon cells to fire action potentials (Greenin-Whitehead et al., 2025).

Moreover, it seems like it is an artificial way to do an un-nested statistics. Similarly, if the grasp of what happen is correct, the stats should be nested, multiple comparisons, and repeated measures.

In Figure 4, our original statistics table said “2-way ANOVA” but it should have said “2-way **repeated-measures** ANOVA” (the paired factor is the segment number). We apologise for the oversight. This test shows: (1) a significant main effect of genotype ($p = 0.0278$), (2) a significant interaction between genotype and segment number ($p < 0.0001$), and (3) significant effects of genotype in post-hoc comparisons at segment numbers 6, 7, 8, 9, 10 and 15 ($p < 0.05$, Sidak’s multiple comparisons test). We have added the post-hoc comparisons to Table A3 and added asterisks to the figure to indicate segments with significant differences between control and dTRPA1 flies, and we now mention the 2-way repeated-measures ANOVA in the legend of Figure 4.

• Fig 1B: Kc and APL are not responding in the same way at the beginning of their response, but this is not acknowledged nor quantified. It likely should, because this has implications for the interpretation of following results. Please quantify and show that it is different from baseline (aka define criteria to say when a cell responds) and to each other.

While quantifying these data, we noticed that in the original manuscript, we had neglected to update Fig. 1B (now Fig. 1A) from a previous version that only had $N=5$ (vs. $N=8$ which is the correct number). We apologise for the oversight; we have now updated the figure to the complete data set. (The new data are the last 3 rows in the Fig. 1A section of Dataset S1.) The reviewer may notice that some of the mean traces are slightly shorter in time in the new version; this is because some of the newly added traces were a bit shorter, either because the movie ended prematurely or because there was a motion artefact at the start/end. The traces look qualitatively the same as before and the overall conclusion remains the same: KCs and APL both respond to dTRPA1 activation of KCs with calcium influx; this calcium influx remains high after 30 min of heating; the calcium influx occurs again after cooling and re-heating.

We quantified both the amplitude and dynamics of the response, to cover alternative interpretations of the reviewer’s comment.

First, we interpreted “*show that it [presumably, $\Delta F/F$] is different from baseline (aka define criteria to say when a cell responds)*” to mean: show that the plateau fluorescence is significantly different from the baseline. (Under this interpretation, “when” means “in which data sets can we say that the cells responded significantly”). We measured the mean $\Delta F/F$ during the plateau of the response in the first heat, cooling, and second heating. (Time periods for averaging are the shaded regions in new Fig. 1A and quantification is in new Fig. 1B). The baseline $\Delta F/F$ is 0 by definition because $\Delta F/F = (F-F_0)/F_0$, so we carried out one-sample t-tests (with Holm-Bonferroni multiple comparisons correction) to test if the plateau $\Delta F/F$ was significantly different from 0. All 6 groups were significantly different from baseline.

We compared the magnitudes of the responses by carrying out a repeated-measures 2-way ANOVA (matching across time and cell type) on the plateau $\Delta F/F$ values. Although APL responses were higher than KC responses (main effect of cell type, $p = 0.003$), this is not biologically meaningful, because in this experiment, APL and KCs express different calcium sensors that have different dynamic ranges (GCaMP6f vs. jRGECO1a). Even if we repeated the experiment with KCs expressing GCaMP in some flies and APL expressing GCaMP in different flies, comparing the magnitude of GCaMP $\Delta F/F$ between cell types would not be biologically meaningful because the two cell types would likely have different expression levels of GCaMP and other calcium buffers, different calcium dynamics (resting $[Ca^{2+}]$, calcium extrusion/sequestration), and different relative localizations of GCaMP and calcium channels.

In the KC traces, the plateau in the first heating response is a bit lower than the plateau in the cooling response, while the reverse is true for the APL traces. However, in the repeated-measures 2-way ANOVA, there is no significant interaction between time (first rise vs. fall vs. second rise) and cell type (KCs vs. APL). In any case, we would not wish to draw major conclusions from this quantification, given that jRGECO1a and GCaMP6f could have different bleaching dynamics during the long recording.

Second, we considered that when the reviewer wrote, “when a cell responds”, they might have been referring to the response onset dynamics, i.e., how fast KCs versus APL respond. However, we cannot define fair criteria to compare when the signal first rises above baseline in KCs vs. APL - the normal way to do this would be something like a z-score based on the baseline noise (standard deviation over time at baseline), where one could say that time of the response is when the change in fluorescence first exceeds, say, 2x the baseline noise. This would not be a fair comparison between KCs and APL because the signal in APL is much lower (being only a single neuron, compared to 2000 KCs), so the noise in fluorescence (relative to signal) is correspondingly higher. As this difference in noise is entirely an artefact of the measurement method, it wouldn't give a meaningful comparison of response latency.

If we consider the overall shape of the response rather than trying to quantify time of first response - the response dynamics are qualitatively very similar, which can be seen by normalizing and overlaying the two average traces:

We quantified this by measuring how long it takes each trace to reach 50% of the plateau value calculated above (the values in Fig. 1B). This “50%-latency” did not differ between APL and KCs (p

= 0.627, paired t-test) and indeed the 50%-latencies were highly correlated between APL and KCs ($r = 0.870$, $p = 0.0025$)

We present the quantification here but have not included it in the manuscript because we don't think it adds much, since it is already visually clear from the average traces in Fig. 1A that the KC and APL responses have similar dynamics.

It is unsurprising that we did not detect any differences in response latencies, since these traces were on the timescale of minutes, while it can be seen from the 5-second odour responses in Figs. 2, 3, A1, A2 that APL and KCs respond essentially simultaneously within the temporal resolution of calcium imaging (note in Fig. 2B, the reason the APL traces appear to rise slightly earlier is that the window for moving-average smoothing was 1 s for APL traces and 0.2 s for KC traces).

• While the table summarizing the statical analysis is useful, the statistical approach is not consistent in paper, especially the fact that repeated measures and paired tests are not done throughout when needed (see fig 1F, 4E)

We have double-checked our statistics and are confident that we have applied repeated measures and paired tests where appropriate. Fig. 1F is not a repeated-measures experiment: the brains are fixed and stained immediately at each time point (0 min, 15 min, 6 h, or 24 h at 31 °C), so the data points at different time points are from different brains and therefore are not paired. We have clarified the repeated-measures ANOVA for Fig. 4D (see point above). Fig. 4E should not use a paired test as the Control and dTRPA1 flies are different samples.

• Lacking the necessary computational skill to properly comment on the model, the only thing to add here is that it is a shame not to use it to make predictions, for instance of what would happen for more protracted / more naturalistic types of stimuli. If feasible (and mind you, it may not be) this could tie in very well with the evolutionary aspects to natural vs unnatural perturbations presented in the discussion (lines 355 onwards).

We have made more explicit a prediction that was implicit in the original manuscript: in the model, a small drop in excitatory synaptic input (or small rise in thresholds) can suffice to silence Kenyon cells. This predicts that if future studies look for, say, changes in excitatory synaptic input in pre-activated Kenyon cells, the magnitude of the change might not be very large (even in γ Kenyon cells, whose activity is strongly reduced). We added this point to the Discussion (lines 579-583 / 627-631):

Future studies will reveal to what extent reduced dendritic calcium influx arises from changes to ion channels (e.g., reduced voltage-gated calcium channels) vs. reduced

synaptic input (through pre- or post-synaptic mechanisms). **These changes may be relatively modest, given that our model predicts that a small reduction in synaptic input can suffice to strongly reduce Kenyon cell activity (Figure 6).**

We are very interested in the question raised by the reviewer about what would happen for more natural perturbations. However, our model only predicts how KC activity changes when excitation, inhibition and thresholds are altered – it doesn't describe how different perturbations would alter excitation, inhibition and thresholds. Thus, the answer to this question will have to await future experiments.

• In figure 1, the example images shown make it impossible to appreciate that the various genetically expressed things are indeed expressed in the correct cells because everything is extremely zoomed in and not annotated. This is essential to assess the entire paper. Please modify the figure and ideally show control experiments of specificity and penetrance of expression, e.g. with staining.

We have added zoomed out images that show the Kenyon cells and APL in context (new Fig. 1C,D,H). (The APL picture has an inverted colour scale, i.e. black = bright fluorescence, because it is hard to see the main neurite from the soma to the calyx in the regular colour scale. This follows the example of (Bonheur et al., 2023) and many other *Drosophila* neuroanatomy papers.)

Specificity: Although the GAL4 drivers we used are not perfectly specific to Kenyon cells and APL, it is obvious from the anatomy which cells are the correct ones. We added 2 sentences in the main text to clarify how we knew we were quantifying CRTC localization on the correct cells:

We expressed CRTC::GFP in Kenyon cells using the driver OK107-GAL4; Kenyon cell somata could be clearly identified as the small cells clustered around the calyx (Figure 1C,D) (line 366 / 407)

And

We measured CRTC::GFP localization in APL using the driver VT43924-GAL4.2; while some non-APL neurons were labelled, APL could be clearly recognized as a large cell body on the lateral edge of the midbrain, with a thick neurite extending toward the calyx, which was filled with APL's characteristic reticulated neurites (Figure 1H) [compare to, e.g., (Liu & Davis, 2008; Lin et al., 2014)]. (line 376 / 420)

Penetrance: The APL neuron was labeled in every hemisphere examined (there is only one APL neuron per hemisphere) [now noted at line 380 / 424]. The OK107-GAL4 driver has been widely characterized as labelling all Kenyon cells (Aso et al., 2009, Greenin-Whitehead et al., 2025), but to confirm that these results using UAS-GFP reporters also extend to UAS-CRTC::GFP, we counted the number of Kenyon cells (identified by the above characteristics) labeled by OK107-GAL4 driving CRTC::GFP, in a subsample of our images (control and TRPA flies kept at 31°C for 24 h; N=6,6). ~2000 Kenyon cells were labeled in both (new Fig. 1G, lines 371-375 / 413-419).

• General point about figures: ideally, they should all be as clear and easy to follow as fig 5 with schematic of experimental design, example traces, and quantification of such traces. For example, Figs. 1a and 2g present the same information, so from a clarity and storytelling prospective it is unclear why they are not combined into a single figure.

We presume the reviewer meant Fig. 1G and 2A. We understand the reviewer's point (indeed, in an earlier version of the manuscript, we put the data from Fig. 1G-H in Fig. 2 instead), but we felt it was more important to have each figure make a single cohesive point. From that point of view, we felt that the data in Fig 1G-H fits better with the APL>CRTC::GFP data, in showing that APL is less responsive to artificial TRPA activation of Kenyon cells - whereas Fig. 2's point is that APL is also less responsive to Kenyon cells with more naturalistic odour responses.

In addition, Fig. 1a is not a schematic of the experiment but rather a hypothesis for the entire study, which makes its placement within an already complex figure confusing.

At Reviewer 1's recommendation, we have made our interpretation and framing more conservative to make the paper less focussed on cell-specific vs. network-level adaptation, so Fig. 1A is no longer relevant. Thus, we have removed it. We copied the schematic from Fig 1G to new Fig 1A (was 1B) to clarify which cells are expressing what. We added schematics illustrating Crtc-GFP to panels 1C and 1H.

Furthermore, Fig. 2 does not include a single example trace to accompany the quantification; all traces are instead placed in the supplementary material. Given this, it is difficult for the reader to understand exactly what is being quantified in Fig. 2. At minimum, a representative example trace should be included in the main figure, with the remaining examples left in the supplementary material.

We thank the reviewer for this suggestion to clarify our presentation. We now include the raw traces from which the example scatter plot was derived as a new panel 2B, with a shaded region to indicate the period over which the mean $\Delta F/F$ is calculated. We also added "quantified the responses as the mean $\Delta F/F$ during the 5 s odour stimulus" to the main text at line 394 / 438.

Minor comments:

- The introduction feels too long and even a bit patronizing. While it is understandable that the authors want to set the scene for what are unexpected results later on, they may want to rein it in a little.

We have shortened the Introduction from 722 to 605 words. We hope this more condensed version feels less didactic.

Instead, they could use the save words to help the non drosophilologist understand better the various genetic lines use as well as the temperature manipulation that is really not common in mammalian work.

We added a bit more text about the temperature manipulation:

We artificially activated Kenyon cells using the heat-activated cation channel dTRPA1, which is normally expressed in *Drosophila* heat-sensing neurons (Hamada et al., 2008). Ectopically expressing dTRPA1 in Kenyon cells allowed us to control their activity by raising the ambient temperature. This thermogenetic tool produces more sustained neuronal activation than channelrhodopsin (Pulver et al., 2009) and has previously been used to induce homeostatic plasticity (Oswald et al., 2018; Apostolopoulou & Lin, 2020). In agreement with previous results (Lin et al., 2014; Apostolopoulou & Lin, 2020; Amin et al.,

2020), raising the temperature to ~31 °C increased intracellular Ca²⁺ in both Kenyon cells (direct activation) and APL (excited by Kenyon cells), as measured in the calyx, where Kenyon cell dendrites reside (Figure 1A,B). We showed previously that in flies without dTRPA1 in Kenyon cells, raising the temperature during the recording does not increase baseline APL Ca²⁺ influx (Lin et al., 2014; Amin et al., 2020), **confirming the effect is due to opening dTRPA1 rather than heating alone.** [line 339-352 / 379-392]

We also made some clarifying edits about fly genetics:

In order to measure APL's sensitivity to Kenyon cell input, we simultaneously recorded from Kenyon cells and APL. **We used the binary expression systems GAL4/UAS and LexA/lexAop (Venken et al., 2011) to express** the green calcium indicator GCaMP6f (Chen et al., 2013) in APL using **the driver** VT43924-GAL4.2 (Amin et al., 2020) and the red calcium indicator jRGECO1a (Dana et al., 2016) in Kenyon cells using **the driver** MB247-LexA (Pitman et al., 2011). [line 334-339 / 374-379]

And:

We tagged endogenous Para with GFP specifically in Kenyon cells using the FlpTag system (Fendl et al., 2020) (Figure 4A), **in which GFP is spliced into Para only in cells expressing FLP recombinase (here, Kenyon cells expressed FLP driven by mb247-LexA).** **In this *para*^{FlpTag} allele,** the FlpTag cassette is inserted into a location that labels >98% of *para* transcripts (Ravenscroft et al., 2020), so almost all isoforms of Para should be labeled **by this *para*^{FlpTag} allele.** [line 452-456 / 500-505]

- Along the same lines, please explain more clearly why the various regions they are recording from are relevant/different to contextualize both fig 2 and 3 for non-drosophila experts.

We have edited the text to read:

We recorded odor responses in the different anatomical regions of the mushroom body: the calyx (Kenyon cell dendrites) and the axonal lobes, formed of the axons of the three main subtypes of Kenyon cells, αβ (forming the α and β lobes), α'β' (α' and β' lobes) and γ (just the γ lobe); the three subtypes differ in their electrical properties (Turner et al., 2008; Inada et al., 2017; Groschner et al., 2018) and their role in different phases of memory (Güven-Ozkan & Davis, 2014). [line 396-401 / 440-446]

- Why using apple cider vinegar from a supermarket together with a panel of monomolecular odours from sigma? Was there a hypothesis of seeing different things? Moreover, why are the monomolecular odours naturalistic?

We didn't have a specific hypothesis of seeing different things; we simply wanted to use a range of different odours including both "food" odours and odours that are widely used in the fly olfaction field. Isoamyl acetate and ethyl butyrate are found in fruit (to humans, they smell like banana and pineapple, respectively). Delta-decalactone was used previously as an odour that activates relatively few olfactory receptor neurons (Lin et al., 2014; Apostolopoulou & Lin, 2020), while 3-octanol and 4-methylcyclohexanol are widely used for olfactory learning experiments (Tully & Quinn, 1985).

- Please annotate images on 4b to orient the reader. While colour choice is commendable since it caters for colour blind colleagues, green and purple merge into massive white smears that looks like noise instead of the interesting signal.

We have added a green box around “MB>ParaFlpTag”, a magenta box around “MB>dsRed”, and a white box with a black border around “merge”. We also added labels for the calyx, peduncle and lobe on one panel.

- Please add figure legend for A1 and A2 instead of just mentioning them in the legend of figure 2.

We have provided the legends for Fig. A1 and A2. We also shrank them down a bit so they will fit in portrait mode.

- Discussion, line 343: please check the analogy with air conditioning and heating, it does not seem to mirror what you are saying for excitation and inhibition.

We removed the analogy (line 588 / 636).

References

- Amin H, Apostolopoulou AA, Suárez-Grimalt R, Vrontou E & Lin AC (2020). Localized inhibition in the *Drosophila* mushroom body. *Elife* 9, e56954.
- Apostolopoulou AA & Lin AC (2020). Mechanisms underlying homeostatic plasticity in the *Drosophila* mushroom body in vivo. *PNAS* 117, 16606–16615.
- Bonheur M, Swartz KJ, Metcalf MG, Wen X, Zhukovskaya A, Mehta A, Connors KE, Barasch JG, Jamieson AR, Martin KC, Axel R & Hattori D (2023). A rapid and bidirectional reporter of neural activity reveals neural correlates of social behaviors in *Drosophila*. *Nat Neurosci* 26, 1295–1307.
- Christiansen F, Zube C, Andlauer TFM, Wichmann C, Fouquet W, Oswald D, Mertel S, Leiss F, Tavosanis G, Luna AJF, Fiala A & Sigrist SJ (2011). Presynapses in Kenyon Cell Dendrites in the Mushroom Body Calyx of *Drosophila*. *J Neurosci* 31, 9696–9707.
- Lin AC, Bygrave AM, de Calignon A, Lee T & Miesenböck G (2014). Sparse, decorrelated odor coding in the mushroom body enhances learned odor discrimination. *Nat Neurosci* 17, 559–568.
- Tully T & Quinn WG (1985). Classical conditioning and retention in normal and mutant *Drosophila melanogaster*. *J Comp Physiol (A)* 157, 263–277.

Dear Dr Lin,

Re: JP-RP-2026-290394R1 "Conflicting adaptations in an inhibitory feedback circuit" by Gregor A Bergmann, Melissa W Tan, Katie Greenin-Whitehead, Philippe J Fischer, Thomas C Cozens, and Andrew C Lin

We are pleased to tell you that your paper has been accepted for publication in The Journal of Physiology.

Yours sincerely,

Nathan Schoppa
Senior Editor
The Journal of Physiology

IMPORTANT POINTS TO NOTE FOLLOWING ACCEPTANCE OF YOUR PAPER:

- **IMPORTANT NOTICE ABOUT OPEN ACCESS:** To assist authors whose funding agencies mandate immediate public access to published research findings, The Journal of Physiology allows authors to pay an Open Access (OA) fee to have their papers made freely available immediately on publication.

- You can help your research get the attention it deserves! Check out Wiley's free Promotion Guide for best-practice recommendations for promoting your work at: www.wileyauthors.com/eeo/guide. You can learn more about Wiley Editing Services which offers professional video, design, and writing services to create shareable video abstracts, infographics, conference posters, lay summaries, and research news stories for your research at: www.wileyauthors.com/eeo/promotion.

- If you would like to receive our 'Research Roundup', a monthly newsletter highlighting the cutting-edge research published in The Physiological Society's family of journals (The Journal of Physiology, Experimental Physiology, Physiological Reports, The Journal of Nutritional Physiology and The Journal of Precision Medicine: Health and Disease), please click this link, fill in your name and email address and select 'Research Roundup':
<https://www.physoc.org/journals-and-media/membernews>

EDITOR COMMENTS

Thank you for sending in your revised manuscript. Both of the original reviewers were quite happy with the changes that you have made. They are convinced by your results and believe that they will be influential in the field of homeostatic plasticity. Also, you responded satisfactorily to a few additional points that I made as an editor. Your manuscript is now acceptable for publication.

REFEREE COMMENTS

Referee #1:

The authors have fully addressed all of my concerns. Congratulations on a really nice piece of work!

Referee #2:

I thank the authors for the very thorough revision and rebuttal letter. I am fully satisfied with the extra details added for both methods and statistics, and for the improved figures. While I still think that the preambles are a bit much, I fully take their point on going over rather than under to facilitate understanding, and anyway this was a stylistic issue.

Overall, I think that the paper is substantially improved and I congratulate them on completing such a thought-provoking study.